# A plant RNA virus inhibits NPR1 sumoylation and subverts NPR1-mediated plant immunity

Jiahui Liu[1,7], Xiaoyun Wu[1,2,7], Yue Fang[1], Ye Liu[1], Esther Oreofe Bello[1], Yong Li[3], Ruyi Xiong[4,6], Yinzi Li[4], Zheng Qing Fu[5], Aiming Wang [4] & Xiaofei Cheng [1,2] ✉

NONEXPRESSER OF PATHOGENESIS-RELATED GENES 1 (NPR1) is the master regulator of salicylic acid-mediated basal and systemic acquired resistance in plants. Here, we report that NPR1 plays a pivotal role in restricting compatible infection by turnip mosaic virus, a member of the largest plant RNA virus genus *Potyvirus*, and that such resistance is counteracted by NUCLEAR INCLUSION B (NIb), the viral RNA-dependent RNA polymerase. We demonstrate that NIb binds to the SUMO-interacting motif 3 (SIM3) of NPR1 to prevent SUMO3 interaction and sumoylation, while sumoylation of NIb by SUMO3 is not essential but can intensify the NIb–NPR1 interaction. We discover that the interaction also impedes the phosphorylation of NPR1 at Ser11/Ser15. Moreover, we show that targeting NPR1 SIM3 is a conserved ability of NIb from diverse potyviruses. These data reveal a molecular "arms race" by which potyviruses deploy NIb to suppress NPR1-mediated resistance through disrupting NPR1 sumoylation.

Plants are continually challenged by various phytopathogens, e.g., viruses, bacteria, fungi, nematodes, and oomycetes. To confront pathogen attacks, plants have evolved multilayered defense mechanisms, such as preformed physical barriers, toxic compounds, RNA silencing, nonhost resistance, and innate immunity[1]. Innate immunity, which is further divided into pathogen-associated molecular pattern (PAMP)-triggered immunity (PTI) and effector-triggered immunity (ETI), plays critical roles in the "arms race" between plants and pathogens[2]. PTI is triggered by the perception of PAMPs via cell surface–localized pattern recognition receptors (PRRs), whereas ETI is activated by the recognition of pathogenic effectors through intracellular nucleotide-binding leucine-rich repeat receptors (NLRs)[3]. The perception of PAMPs or effectors causes a significant conformational alternation of the receptors and results in a series of downstream signaling events that lead to resistance responses, e.g., the expression of defense-related genes, the synthesis and deposition of callose at the plasmodesmata (PD), strengthening of the cell wall, and even a hypersensitive response (HR)[4]. Recent studies have shown that PTI and

ETI work synergistically by potentiating each other to achieve stronger immune responses[5,6].

Activation of PTI and ETI also induces resistance in distal tissues, a phenomenon known as systemic acquired resistance (SAR). SAR is mediated by the defense hormone salicylic acid (SA) through the function of NONEXPRESSER OF PATHOGENESIS-RELATED GENES 1 (NPR1)[7,8]. In Arabidopsis, NPR1 is present in the cytoplasm as tetramers via intermolecular disulfide bonds under steady-state conditions[9]. SA rapidly accumulates upon the activation of PTI and ETI from both de novo biosynthesis and hydrolysis of inactivated forms[10]. The increased SA level alters the cellular redox state, which causes monomerization of cytoplasmic NPR1 via thioredoxins[11]. NPR1 is then phosphorylated by SNF1-RELATED PROTEIN KINASE 2.8 (SnRK2.8) at Ser589 and possibly Thr373, and these phosphorylations promote NPR1's nuclear entry[12]. In the nucleus, NPR1 is further sumoylated by SMALL UBIQUITIN-LIKE MODIFIER 3 (SUMO3) and phosphorylated at Ser11/Ser15; then, it reprograms overall transcription via basic region/leucine zipper motif (bZIP) and WRKY transcription factors[13,14]. NPR1 is also

[1]College of Plant Protection, Northeast Agricultural University, 150030 Harbin, Heilongjiang, China. [2]Key Laboratory of Germplasm Enhancement, Physiology and Ecology of Food Crops in Cold Region of Chinese Education Ministry, Northeast Agricultural University, 150030 Harbin, Heilongjiang, China. [3]College of Life Science, Northeast Agricultural University, 150030 Harbin, Heilongjiang, China. [4]London Research and Development Centre, Agriculture and Agri-Food Canada, London N5V 4T3 ON, Canada. [5]Department of Biological Sciences, University of South Carolina, Columbia, SC 29208, USA. [6]Present address: A&L Canada Laboratories Lnc., London N5V 3P5 ON, Canada. [7]These authors contributed equally: Jiahui Liu, Xiaoyun Wu. ✉e-mail: xfcheng@neau.edu.cn

subjected to ubiquitin modification through Cullin-RING E3 ligase and UBIQUITIN E4 LIGASE/MUTANT, SNC1-ENHANCING 3 (UBE4/MUSE3) and is degraded through the 26S ubiquitin–proteasome system in the nucleus[15]. Interestingly, ubiquitination is also required for the full transcriptional activity of NPR1 during SAR[16]. In addition, NPR1 directly participates in ETI by forming SA-induced NPR1 condensates (SINCs) to modulate the turnover of stress response proteins in cells surrounding the HR sites to promote their survival[17].

Viral pathogens are not generally viewed as PAMP-encoding intruders, since recognition of obligate intracellular parasites by extracellular PRRs is counterintuitive[18]. It is believed that ETI and RNA silencing are the two principle antiviral mechanisms in plants. However, how plants respond to infection by compatible viruses that spread systemically and cause disease is less well understood. Transcriptomic and proteomic studies have revealed extensive transcriptional reprogramming during compatible plant–virus interactions, including upregulation of a group of defense-related genes[19]. Direct treatment with SA or its analog in wild-type (WT) plants but not *npr1* mutants induces resistance to various phytopathogens, including viruses[20]. NPR1-knockout or NPR1-knockdown plants, SA biogenesis-deficient mutants, and transgenic plants expressing a salicylate hydroxylase gene (*NahG*) all show enhanced susceptibility to infection by adapted viruses[21–25]. These findings imply that the SA signaling cassette may play a role in compatible plant–virus interactions. However, how the SA signaling cassette activated, its contribution to compatible plant–virus interactions, and how plant viruses manipulate SA-mediated plant defenses are poorly understood.

The family *Potyviridae* includes more than 30% of known plant-infecting RNA viruses, which are divided into at least 12 definitive genera and 3 unassigned species[26]. *Potyvirus* is the largest genus in the family, and includes many agriculturally important viruses, such as potato virus Y (PVY), turnip mosaic virus (TuMV), soybean mosaic virus (SMV), and papaya ringspot virus (PRSV)[27]. The genome of typical potyviruses consists of a positive-sense single-stranded RNA (+ssRNA) of ~9.6 K nucleotides (nt) that contains a single open reading frame (ORF) encoding a large polypeptide of ~350 kDa. In addition, a unique polymerase slippage motif within the *P3* cistron enables the expression of an additional short polypeptide[28]. These two polypeptides are proteolytically processed by three viral proteases into 11 mature proteins and many precursors[26]. Although the replication of potyviruses takes place in the cytoplasm, several viral proteins, including NUCLEAR INCLUSION B (NIb), the viral RNA-dependent RNA polymerase (RdRp)[29], localizes primarily in the nucleus for unknown reasons[30]. Previously, we reported that TuMV-encoded NIb interacts with the components of Arabidopsis sumoylation machinery, e.g., SUMO-CONJUGATING ENZYME 1 (SCE1) and SUMO3, in the nucleus and displays sumoylation-dependent immunosuppressive activity[31,32], suggesting that nucleus-localized NIb may have an important function in inhibiting plant immunity.

In this work, we further demonstrate that the SA signaling cassette plays a critical role in compatible plant–virus interactions and that NIb directly targets NPR1 to suppress such resistance in the nucleus.

## Results

### NIb interacts with NPR1 and targets its SUMO-interacting motif 3

We have previously found that TuMV-encoded NIb interacts with and is sumoylated by SUMO3 in Arabidopsis to promote virus infection[32]. SUMO3 also positively regulates Arabidopsis immunity, possibly by interacting with or modifying its components, such as NPR1[33,34]. Since NIb displays sumoylation-dependent immunosuppressive activity[32], we suspected that NIb may directly interact with SUMO3 substrates to compete for SUMO3. To test this idea, we performed a bimolecular fluorescence complementation (BiFC) assay between NIb and known SUMO3 substrates, e.g., NPR1, BRAMHMA (BRM), CYTOKININ RESPONSE FACTOR 6 (CRF6), several TEOSINTE BRANCHED1/

CYCLOIDEA/PROLIFERATING CELL FACTOR (TCP) transcription factors, SQUAMOSA PROMOTER BINDING PROTEIN-LIKE 4/FLORAL TRANSITION AT THE MERISTEM 6 (SPL4/FTM6), ETHYLENE RESPONSIVE ELEMENT BINDING FACTOR 5 (ERF5), and FLOWERING BHLH 3 (FBH3)[34–36]. The signal of yellow fluorescent protein (YFP) was recorded in *Nicotiana benthamiana* epidermal cells expressing the C-terminal segment of YFP (YC)-fused NIb (NIb-YC) and the N-terminal segment of YFP (YN)-fused Arabidopsis NPR1, TCP3, TCP4, TCP23, BRM, SPL4/FTM6, ERF5, FBH3, and CRF6 (Supplementary Fig. 1), indicating that NIb may interact with or in the same complex as all tested SUMO3 substrates. Given the central role of NPR1 in plant immunity and the exclusively nuclear YFP signal from NIb-YC and NPR1-YN (Supplementary Fig. 1), we focused our study on the NIb–NPR1 interaction.

A yeast two-hybrid (Y2H) assay was performed to confirm the interaction between NIb and NPR1 (Fig. 1a and Supplementary Fig. 2a). Yeast cells cotransformed with the GAL4 activation domain (AD) and GAL4 DNA-binding domain-fused NIb (BD-NIb) or AD-NPR1 and BD did not survive on selective medium lacking tryptophan, leucine, histidine, and adenine (-AHLW), while those transformed with AD-NPR1 and BD-fused TGACG-BINDING FACTOR 3 (BD-TGA3), a known NPR1-interacting transcription factor[37], survived on selective medium, confirming the specificity of the system. When AD-NPR1 was coexpressed with BD-NIb or BD-SUMO1, we found that histidine auxotrophy was restored only when NPR1 was cotransformed with NIb but not with SUMO1, which was used as a negative control (Fig. 1a). Moreover, SA had no obvious influence on the interaction between NIb and NPR1 (Supplementary Fig. 2b). We further tested several known dysfunctional NPR1 mutants, e.g., Cys82 to Ala (C82A), Cys156 to Ala (C156A), Cys216 to Ala (C216A), and His334 to Tyr (H334Y). The results showed that NIb also interacted with C82A, C156A, and C216A, but not H334Y (Fig. 1a and Supplementary Fig. 2a). Consistently, YFP signals were also recorded in the nuclei of *N. benthamiana* epidermal cells expressing NIb-YC and C82A-YN, C156A-YN, or C216A-YN, but not H334Y-YN (Fig. 1b; Supplementary Fig. 2c). We further confirmed the interaction by coimmunoprecipitation (Co-IP). The results showed that both FLAG-4×Myc-tagged NPR1 and SUMO3, which was used as a positive control, were coprecipitated by NIb-YFP (Fig. 1c). We also performed a glutathione S-transferase (GST) pulldown assay using purified proteins from *Escherichia coli*. The results showed that thioredoxin-6×Histidine (Trx-6×His)-tagged NPR1 but not the free Trx-6×His tag was successfully pulled down by GST-NIb but not GST (Fig. 1d). Very recently, NPR1 was also identified in a high-throughput Y2H screening using NIb as the bait[38]. Together, these results clearly indicate that TuMV-encoded NIb physically interacts with Arabidopsis NPR1 in the nucleus.

To determine the residues of NPR1 that are responsible for NIb interactions, we divided NPR1 into three nonoverlapping fragments based on a previous study[7], namely, the N-terminal bric-à-brac, tram track, and broad complex/poxvirus and zinc finger (BTB/POZ) domain (amino acids 1-230), the central ankyrin repeat (ANK) domain (amino acids 231–465), and the C-terminal putative transcriptional activation (TAD) domain that binds SA (amino acids 466-593; Fig. 1e). The Y2H and BiFC results showed that NIb interacted with only the ANK domain (Fig. 1f, g and Supplementary Fig. 3). Since NIb failed to interact with H334Y and exhibits sumoylation-dependent immunosuppressive activity[32], we suspected that the SIM3 (amino acids 345–348) neighboring His334 may be essential for NIb interaction[34]. Indeed, we found that SIM3-mutated NPR1 mutant (sim3) failed to interact with NIb in both Y2H and BiFC assays (Fig. 1f, g and Supplementary Fig. 3). Previous studies have shown that SIM3 and His334 affect NPR1 nucleocytoplasmic partitioning[34]. Transiently expressed NPR1, C82A, C156A, and C216A were located mostly in the nuclei, while sim3 and H334Y were located both in the nuclei and cytoplasm of *N. benthamiana* epidermal cells (Supplementary Fig. 4a–f). Coexpression of NIb had no obvious influence on the subcellular localization of NPR1, H334Y, or

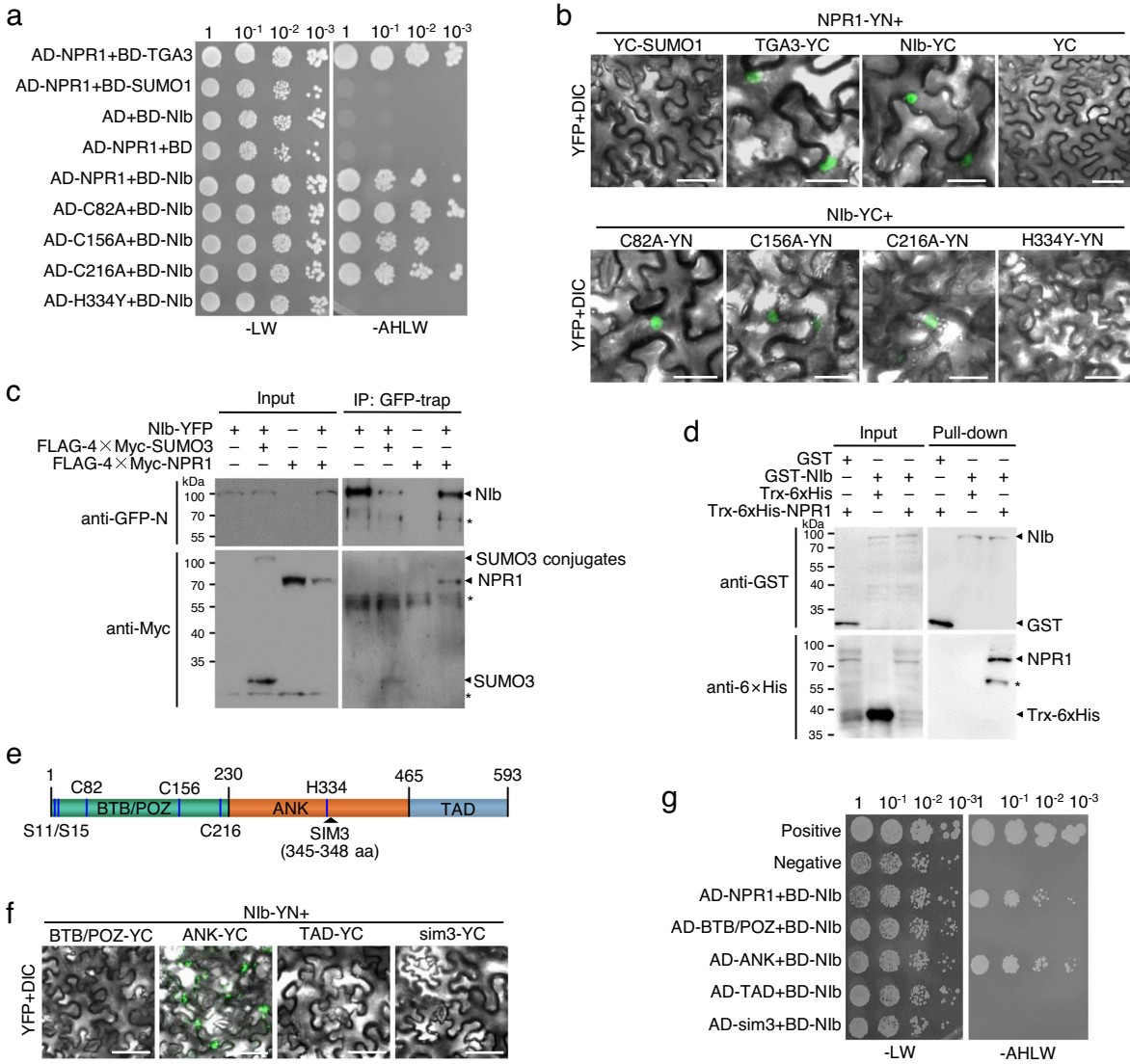

**Fig. 1 | TuMV NIb interacts with Arabidopsis NPR1. a** Growth of serially diluted yeast cells that were transformed with the indicated plasmids on selective medium. **b** Confocal microscopic photographs of *N. benthamiana* epidermal cells that were infiltrated by agrobacteria harboring the indicated plasmids at 2 dpi. Scale bar, 50 µm. The experiment was independently repeated three times with similar results. **c** Co-IP assay to test the interaction between NIb and NPR1 or SUMO3. FLAG-4×Myc or YFP-tagged proteins were expressed in *N. benthamiana* leaves by agroinfiltration, immunoprecipitated with GFP-trap agarose at 2 dpi, and detected with anti-GFP N-terminal (anti-GFP-N) or anti-Myc antibodies, respectively. Asterisks indicate nonspecific bands. The experiment was independently repeated twice with similar results. **d** In vitro binding assay with purified proteins from

*E. coli*. GST or GST-NIb was used as matrix-bound bait to bind TrxA-6×His or TrxA-6×His-NPR1. The asterisk indicates degraded NPR1. The experiment was independently repeated twice with similar results. **e** Schematic representation of NPR1. Numbers represent amino acid positions of domain boundaries. Ser11/Ser15, Cys82, Cys156, Cys216, His334, and SIM3 are also indicated. **f** Confocal microscopic photographs of *N. benthamiana* epidermal cells expressing NIb-YN and YC-tagged NPR1-truncated mutants or sim3 at 2 dpi. Scale bar, 50 µm. The experiment was independently repeated three times with similar results. **g** Growth of serially diluted yeast cells that were transformed with BD-NIb and AD-tagged NPR1-truncated mutants or sim3 on selective medium.

sim3 (Supplementary Fig. 4g–i), which ruled out the possibility that the failed interaction between NIb and sim3 or H334Y was due to absence from the nucleus. Together, the above results suggest that NIb physically interacts with NPR1 and binds to the region within the ankyrin repeat domain of NPR1, which includes His334 and SIM3.

**NPR1-dependent immune responses restrict TuMV infection**
We then tried to dissect the biological function of the NIb–NPR1 interaction in TuMV infection using three *npr1* mutants (*npr1-1*, *npr1-0*, and *npr1-6*). *npr1-1* harbors H334Y[7]; *npr1-0* (SALK_204100) is a null mutant that has a T-DNA insertion in the first exon of the *NPR1* gene[16]; and *npr1-6* (SAIL_708_F09) contains a T-DNA insertion in the third exons of the *NPR1* gene, which allows the expression of amino acids

1-466 of NPR1 (NPR1ΔC)[39]. NPR1ΔC lacks the nuclear localization signal[40]. As a result, transiently expressed NPR1ΔC-YFP was located mostly in the cytoplasm of *N. benthamiana* epidermal cells (Supplementary Fig. 5a). Three-week-old WT Arabidopsis ecotype Col-0, *npr1-1*, *npr1-0*, and *npr1-6* seedlings were agro-inoculated with TuMV-GFP, a TuMV infectious clone expressing a free green fluorescent protein (GFP) between the P1 and HC-Pro cistrons, to directly visualize virus infection[40]. At 14 days post-inoculation (dpi), we found that the ratios of TuMV-infected leaf area to total rosette leaf area of *npr1-1*, *npr1-0*, and *npr1-6* were significantly higher than that of the WT, as indicated by the GFP fluorescence from TuMV-GFP (Fig. 2a, b). Reverse transcription and quantitative PCR (RT–qPCR) showed that the three *npr1* mutants accumulated more than 1.5-fold more viral genomic RNAs

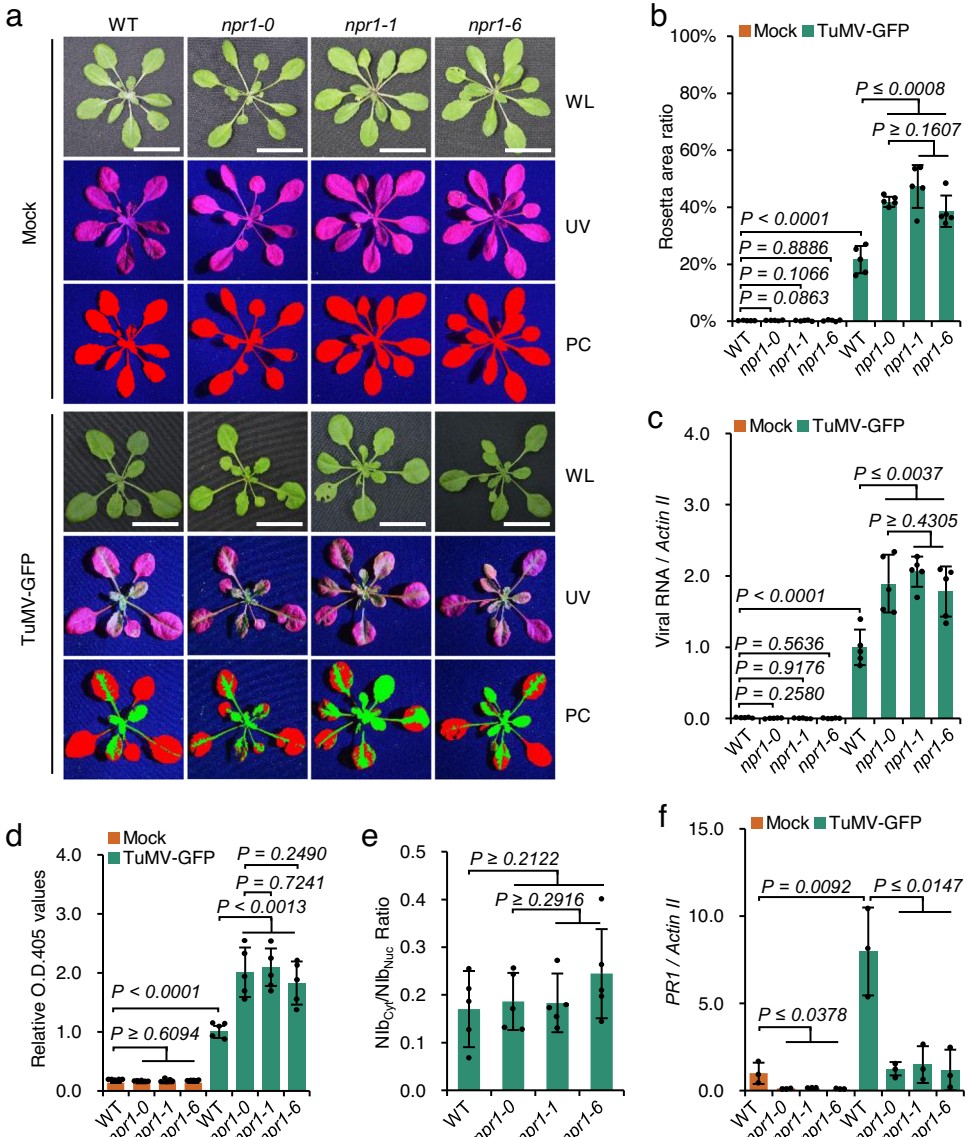

**Fig. 2 | *npr1* mutants are hypersusceptible to TuMV infection. a** Phenotypes of WT plants and *npr1* mutants agroinfiltrated with infiltration buffer (mock) or agrobacteria harboring TuMV-GFP under white light (WL) and ultraviolet light (UV) at 14 dpi. Red and green colors in the pseudocolor (PC) panel refer to virus-free and virus-infected areas, respectively. **b** Bar chart showing the ratios of TuMV-infected to the total leaf area of WT plants and *npr1* mutants at 14 dpi (*n* = 5). **c** Bar chart of the relative levels of the viral genome in WT plants and *npr1* mutants at 14 dpi. RT–qPCR was performed with *Actin II* as the internal control, and the viral genome in the WT was normalized to 1 (*n* = 5). **d** Bar chart of the relative virion amounts in WT plants and *npr1* mutants at 14 dpi. ELISA reads were taken after 1 h of substrate hydrolysis (*n* = 5). **e** Bar chart of the ratios of cytoplasmic NIb (NIb$_{Cyt}$) to nuclear NIb (NIb$_{Nuc}$) in WT or *npr1* mutants (*n* = 5). **f** Bar chart of the relative *PR1* expression level in mock- or TuMV-infected WT plants and *npr1* mutants at 48 hpi (*n* = 3). RT–qPCR was performed with *Actin II* as the internal control, and the *PR1* level in mock-infected WT plants was normalized to 1. Data are presented as mean values ± SD. Statistical analyses were performed using Two-tailed Student's *t* test. Source data are provided as a Source Data file.

than the WT (Fig. 2c). Enzyme-linked immunosorbent assay (ELISA) showed that the accumulation of viral particles in three *npr1* mutants was increased 1.8- to 2.1-fold higher than that in WT plants (Fig. 2d), indicating that functional NPR1 is required for restriction of TuMV infection. Moreover, the expression of H334Y in *npr1-1* or NPR1ΔC in *npr1-6* had no additional effect in inhibiting TuMV proliferation compared to that in *npr1-0* (Fig. 2a–d), indicating that NPR1 has little or no ability to directly inhibit NIb RdRp activity or virus proliferation.

Previously, we found that the nucleocytoplasmic shuttling of NIb, which is essential for robust virus proliferation, is affected by SUMO3-mediated sumoylation[32]. Transient expression assays showed that NPR1, sim3, and H334Y had no obvious influence on the subcellular localization of NIb in *N. benthamiana* epidermal cells (Supplementary Fig. 5b, c). Interestingly, the subcellular localization of NIb also was not

affected by NPR1ΔC, which retained full ability to interact with NIb (Supplementary Fig. 5d–f). To further confirm this hypothesis, we compared the nucleocytoplasmic partitioning of NIb in virus-infected WT, *npr1-1*, *npr1-0*, and *npr1-6* seedlings. We found that only ~15.8% of NIb was located in the cytoplasm in WT seedlings (mean cytoplasm/nucleus ratio = 17.03%; Fig. 2e and Supplementary Fig. 6), confirming that the majority of NIb was located in the nucleus[30]. Interestingly, there was no significant difference in the nucleocytoplasmic partitioning of NIb in *npr1-0*, *npr1-1*, and *npr1-6* compared to that in the WT (Fig. 2e and Supplementary Fig. 6), suggesting that NPR1 does not affect NIb nucleocytoplasmic partitioning. We suspected that the varied susceptibilities of WT plants and *npr1* mutants to TuMV were due to NPR1-dependent immune responses. We thus compared the immune responses of these mutants to TuMV infection by monitoring

the expression of *PATHOGENESIS-RELATED 1* (*PR1*), a marker gene of plant immunity[41]. Four-week-old seedlings of the WT and the three *npr1* mutant strains were mechanically inoculated with TuMV-GFP to avoid the stimulation of agrobacteria. We found that TuMV-GFP infection induced the expression of *PR1* in both Col-0 and *npr1* mutants compared with mock-infected plants at 2 dpi (Fig. 2f and Supplementary Fig. 7a, b). The expression of *PR1* was upregulated by more than 5 times in TuMV-infected WT plants compared with mock-infected WT plants, while the expression of *PR1* in *npr1-1*, *npr1-0*, and *npr1-6* seedlings rose from below the detectable level to a comparable level in mock-infected WT plants (Fig. 2f and Supplementary Fig. 7a, b), indicating that NPR1 plays a dominant role in inducing *PR1* expression during a compatible virus–host interaction. We further compared the sumoylation and phosphorylation of NPR1 before and after TuMV infection using a transgenic Arabidopsis line expressing a C-terminal GFP-tagged NPR1 under the cauliflower mosaic virus (CaMV) 35S promoter in the *npr1-1* background (*35S::NPR1-GFP-3*; Supplementary Fig. 12a, b). Results showed that the infection of TuMV caused an evident increase in the level of NPR1 sumoylation and phosphorylation modifications (Supplementary Fig. 7c), indicating that TuMV infection stimulates the SA-NPR1 signaling pathway. Together, the above results indicate that NPR1-dependent immune responses, but not NPR1, inhibit TuMV infection and that NIb interacts with NPR1 to suppress antiviral immunity.

### NIb disturbs the NPR1-SUMO3 interaction and subsequent sumoylation

We suspected that NIb might disrupt the NPR1–SUMO3 interaction or sumoylation as the SIM3 is indispensable for SUMO3-mediated sumoylation of NPR1[34]. To test this idea, competitive BiFC and split-luciferase assays were performed in the presence of NIb or SUMO1. An NIb mutant (NIb$_{sim2}$) in which the second SIM (SIM2) was mutated was also included, since NIb also interacts with and is sumoylated by SUMO3 via this motif[32]. We found that although NPR1 and SUMO3 were expressed at similar levels, the nuclear YFP signal and luciferase activity were significantly decreased by NIb compared with SUMO1 (Fig. 3a, b and Supplementary Fig. 8), implying that NIb interferes with the interaction of NPR1 with SUMO3. Interestingly, the YFP signal and luciferase activity also decreased to a lesser extent in the presence of NIb$_{sim2}$ (Fig. 3a, b and Supplementary Fig. 8), suggesting that NIb–SUMO3 interaction or SUMO3-mediated sumoylation enhances NIb–NPR1 interaction. Indeed, altering SIM2 or the major sumoylation site (Lys409) of NIb greatly reduced its affinity for NPR1 in Y2H and BiFC assays (Supplementary Fig. 9). Notably, NIb$_{sim2|K409R}$ still interacted with NPR1 (Supplementary Fig. 9), indicating that SUMO3 and sumoylation are not essential for NIb to interact with NPR1. To discern NIb interruption of the NPR1–SUMO3 interaction and inhibition of NPR1 sumoylation, competitive BiFC and split-luciferase assays were performed using a sumoylation-defective mutant of SUMO3 (SUMO$\Delta$GG), in which the C-terminal double glycine motif (Gly-Gly) was mutated to Ala. The results showed that the YFP signal and luciferase activity from the interaction between NPR1 and SUMO$\Delta$GG were also weakened by NIb compared with SUMO1 (Supplementary Fig. 10). Consistently, the presence of NIb$_{sim2}$ decreased the nuclear YFP signal and luciferase activity from the NPR1–SUMO3$\Delta$GG interaction to a lesser extent than NIb (Supplementary Fig. 10), suggesting that NIb directly inhibits NPR1–SUMO interaction. To investigate whether NIb can inhibit the sumoylation of NPR1 by SUMO3, we compared the amounts of SUMO3-sumoylated NPR1 in the presence and absence of NIb by Co-IP in a transient expression assay. We found that at similar expression levels of SUMO3 and NPR1, the amount of nonmodified and posttranslationally modified NPR1 was greatly reduced in the presence of NIb (Supplementary Fig. 11a). We further transformed a 17-β-estrogen-induced NIb (*XVE::NIb*) into *35S::NPR1-GFP-3* to produce *XVE::NIb 35S::NPR1-GFP*. Homozygous plants were treated with 1 mM

SA alone or together with 17-β-estrogen to induce the expression of NIb. NPR1 was then immunoprecipitated by GFP-Trap agarose and detected with anti-SUMO3 antibodies. The results showed that the level of sumoylated NPR1 was significantly reduced by the presence of NIb (Fig. 3c). We further analyzed the impact of NIb on the NPR1–TGA3 interaction since sumoylation promotes the interaction[9,34]. A competitive split-luciferase assay showed that the NPR1–TGA3 interaction was also disrupted by NIb (Fig. 3d and Supplementary Fig. 11b).

Transgenic plants expressing a C-terminal GFP-tagged sim3 or NPR1 under the CaMV 35S promoter in the *npr1-1* background (*35S::sim3-GFP* or *35S::NPR1-GFP*) were produced to further evaluate the influence of NIb on NPR1 sumoylation. All homozygous seedlings displayed a normal phenotype similar to that of WT plants under steady-state conditions (Supplementary Fig. 12a). Two lines of *35S::NPR1-GFP* seedlings and one line of *35S::sim3-GFP* seedlings with detectable NPR1-GFP or sim3-GFP were further analyzed (Supplementary Fig. 12b, c). A previously described transgenic line expressing sim3-GFP (*35S::npr1^sim3^*) was also included[34]. TuMV inoculation showed that the ratios of TuMV-infected to total rosette leaves of *35S::NPR1-GFP–3*, sim3-GFP and *35S::npr1^sim3^* plants were comparable to that of the WT at 18 dpi, while *35S::NPR1-GFP–4* seedlings had a slightly reduced ratio of TuMV-infected to total rosette leaves (Fig. 3e and Supplementary Fig. 12d). RT–qPCR and ELISA showed that *35S::NPR1-GFP-4* seedlings accumulated lower levels of viral genomic RNA and virions than WT seedlings, while *35S::NPR1-GFP-3*, *35S::sim3-GFP* and *35S::npr1^sim3^* seedlings accumulated similar levels of viral genomic RNA and virions as WT seedlings (Supplementary Fig. 12e, f). These findings indicate that sim3-GFP and *35S::npr1^sim3^* seedlings may have susceptibility similar to that of WT plants and that *35S::NPR1-GFP-4* seedlings are more resistant than WT seedlings. To role out the influence of RNA silencing induced by the *GFP* sequence fused to NPR1 or sim3, we analyzed the small RNA (sRNA) profiles of these transgenic seedlings by high-throughput sequencing. The results showed that no or less than 1.0 GFP-derived sRNA per million of total sRNAs was detected in *35S::NPR1-GFP-3*, *35S::sim3-GFP* and *35S::npr1^sim3^* (Supplementary Fig. 12g), confirming that RNA silencing is not a major factor affecting susceptibility in *35S::sim3-GFP* and *35S::npr1^sim3^*. Interestingly, *35S::NPR1-GFP-4* had 4.45 GFP-derived reads per million of total sRNAs (Supplementary Fig. 12g), indicating that the extraordinary resistance of *35S::NPR1-GFP-4* may be due to RNA silencing. We thus inoculated the transgenic plants with TuMV-mCh, a TuMV infectious clone that contains an additional mCherry-tagged 6K2 between P1 and HcPro cistrons[42]. All transgenic plants accumulated similar levels of viral genome at 18 dpi as determined by RT–qPCR (Fig. 3f). Moreover, the expression of *PR1* in *35S::NPR1-GFP*, *35S::sim3-GFP*, and *35S::npr1^sim3^* seedlings was upregulated to a level similar to that in the WT at 2 dpi (Fig. 3g), suggesting that TuMV induced comparable levels of immune responses to TuMV in these plants. Taken together, the above results confirm that the binding of NIb inhibits the NPR1–SUMO3 interaction and subsequent sumoylation by SUMO3 to suppress NPR1-mediated antiviral responses.

### NIb also impacts the sumoylation-dependent phosphorylation of NPR1

Sumoylation and phosphorylation at the first NF-κB inhibitor (IκB)-like phosphodegrons (Ser11/Ser15) orchestrate NPR1 activity[34]. We thus replaced the two serine residues with alanine or aspartic acid in NPR1 or sim3 to further analyze the influence of NIb on NPR1 function. Y2H showed no obvious difference between NIb and NPR1 or the phosphorylation-defective mutant (S11/15A) but a significant growth enhancement of yeast cells that had been cotransformed with NIb and the phosphorylation-mimic mutant (S11/15D) (Fig. 4a and Supplementary Fig. 13a). Interestingly, yeast cells transformed with NIb and sim3|S11/15D but not NIb and sim3|S11/15A survived on the selective medium (Fig. 4a). The BiFC assay showed that the YFP fluorescence in

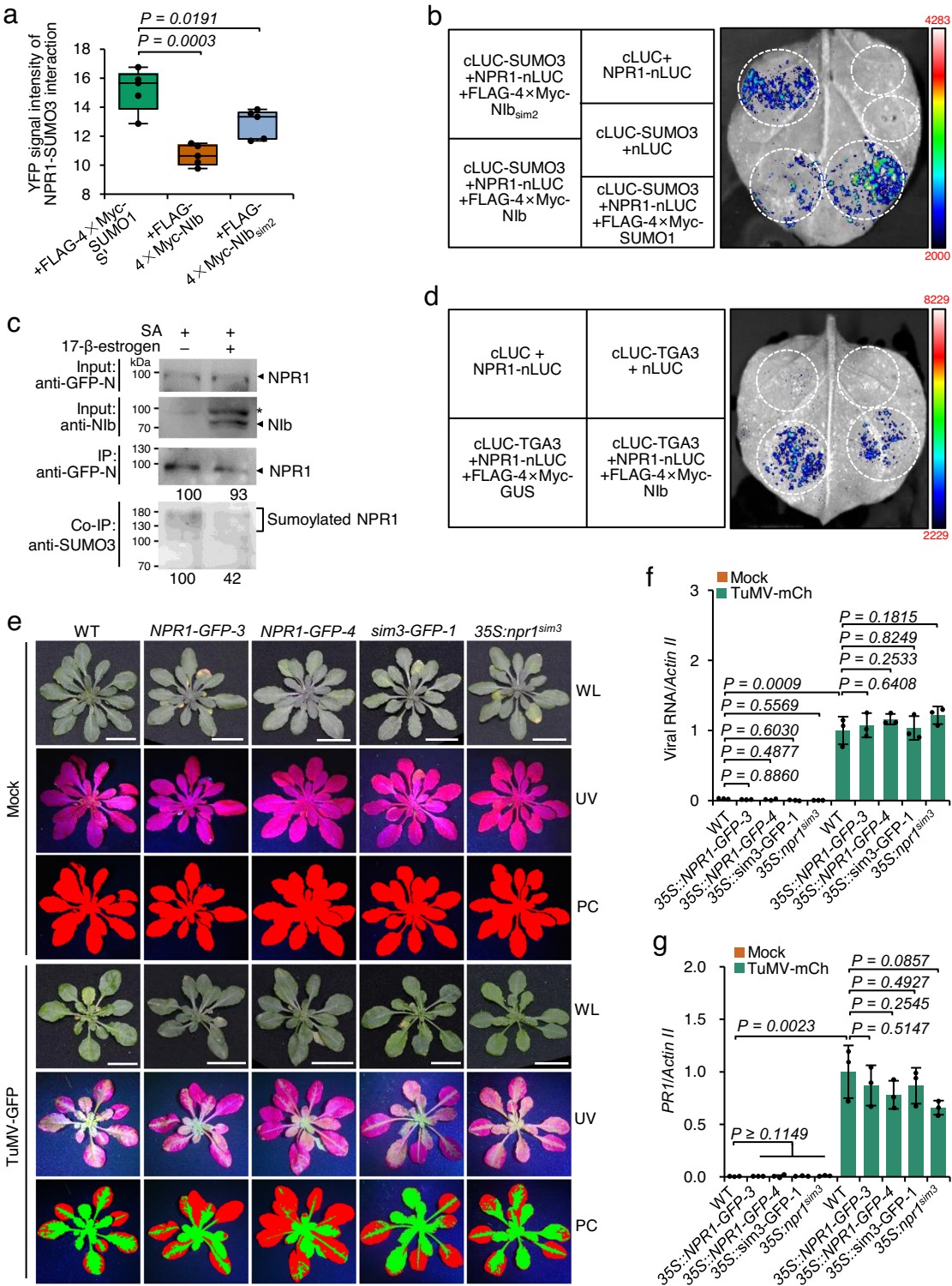

**Fig. 3 | NIb inhibits NPR1 from interacting with and being modified by SUMO3.**
**a** Box and whisker plot with individual data points comparing the nuclear YFP
signal intensity from the NPR1–SUMO3 interaction in the presence of SUMO1, NIb,
or NIb$_{sim2}$. The data are the mean of the nuclear signal intensity of 5 micrographs.
The whisker indicates minimum and maximum, and the box indicates the 25th and
75th percentiles (edges of the box), and median (center line). The original figures
are available in the Figshare repository under [https://doi.org/10.6084/m9.
figshare.23243918]. **b** Influence of SUMO1, NIb, or NIb$_{sim2}$ on the luciferase activity
from the interaction between cLUC-SUMO3 and NPR1-nLUC. The high-low refer-
ence bar shows fluorescence signals, ranging from high (top) to low (bottom).
**c** Immunoblots for NPR1 sumoylation in *XVE::NIb 35S::NPR1-GFP* seedlings after
1 mM SA treatment with or without 2 μM 17-β-estrogen. NPR1, NIb, and SUMO3

were detected with anti-GFP, anti-NIb, and anti-SUMO3 antibodies, respectively.
**d** Competitive split-luciferase assay to evaluate the influence of NIb and GUS on
the NPR1–TGA3 interaction. **e** Phenotypes of mock- and TuMV-infected WT,
*35S::NPR1-GFP* (*NPR1-GFP*), *35S::sim3-GFP* (*sim3-GFP*) and *35S::npr1*$^{sim3}$ seedlings at 18
dpi. **f** Bar chart of the relative TuMV-mCh genome amount in WT and transgenic
plants at 18 dpi (*n* = 3). RT–qPCR was performed with *Actin II* as the internal control,
and the viral genome in TuMV-infected WT plants was normalized to 1. **g** Bar chart
of relative *PR1* expression level in mock or virus-infected WT, *35S::NPR1-GFP*,
*35S::sim3-GFP*, and *35S::npr1*$^{sim3}$ plants at 48 hpi (*n* = 3). The expression of *PR1* in
TuMV-infected WT plants was normalized to 1. Data are presented as mean
values ± SD. Statistical analyses were performed using Two-tailed Student's *t* test.
Source data are provided as a Source Data file.

*N. benthamiana* epidermal cells expressing S11/15D-YC and NIb-YN was also stronger than that in those expressing NPR1-YC and NIb-YN (Fig. 4b and Supplementary Fig. 13b). Moreover, the YFP fluorescence of NIb-YN and sim3|S11/15D-YC was also higher than that of NIb-YN and sim3|S11/15A-YC or sim3-YC (Fig. 4b). Together, these results suggest that phosphorylation of Ser11/Ser15 may cause a conformational change in NPR1 that facilitates NIb binding.

We produced transgenic plants expressing S11/15D-GFP, sim3|S11/15D-GFP, or sim3|S11/15A-GFP under the CaMV 35S promoter in the *npr1-1* background. At least 5 independent lines per construct were obtained. All homozygous *35S::sim3|S11/15D-GFP* and *35S::sim3|S11/15A-GFP* plants with confirmed protein expression had phenotypes similar to those of WT plants, while most lines of *35S::S11/15D-GFP* (5/9) plants displayed autoimmune phenotypes, e.g., a smaller size with curly rosette leaves under steady-state conditions (Fig. 4c and Supplementary Fig. 14a–c). Interestingly, the phenotype severity and *PR1* expression level were positively correlated with the level of S11/15D-GFP (Fig. 4c and Supplementary Fig. 14c, d), confirming that the phosphorylation of Ser11/Ser15 is essential for NPR1 to induce the expression of defense-related genes. Three-week-old transgenic seedlings of two independent lines per construct were agroinfiltrated with TuMV-GFP. At 18 dpi, GFP signals were recorded on both all rosette leaves of WT and *35S::sim3|S11/15A-GFP* seedlings, only on the central rosette leaves of *35S::sim3|S11/15D-GFP* seedlings, but not observed on both lines of *35S::S11/15D-GFP* seedlings (Fig. 4d, e). RT–qPCR and immunoblotting showed that *35S::S11/15D-GFP* seedlings accumulated the lowest viral genome and particle levels, followed by *35S::sim3|S11/15D-GFP* seedlings, while *35S::sim3|S11/15A-GFP* seedlings accumulated viral genome and particle levels comparable to those of WT plants (Supplementary Fig. 15a, b). These results suggest that *35S::S11/15D-GFP* seedlings are highly resistant to TuMV-GFP and *35S::sim3|S11/15D-GFP* seedlings also display considerable resistance to TuMV-GFP compared with that of WT seedlings. We further compared ability of these transgenic plants in inducing immune responses against TuMV infection. The RT-qPCR data showed that the expression of *PR1* in the seedlings of *35S::sim3|S11/15D-GFP* or *35S::sim3|S11/15A-GFP* was similar as that in WT plants. However, the expression of *PR1* was significantly elevated in both lines of *35S::S11/15D-GFP* under steady-state conditions (Supplementary Fig. 15c). After TuMV challenge, *PR1* was upregulated by ~6–10-fold in WT and *35S::sim3|S11/15A-GFP* seedlings, but almost no changed was observed in *35S::S11/15D-GFP* seedlings (Supplementary Fig. 15c), indicating that the immune responses are fully activated in *35S::S11/15D-GFP* seedlings even under steady-state conditions. Interestingly, the expression of *PR1* in *35S::sim3|S11/15D-GFP* seedlings was increased after TuMV challenge to a level comparable to that of *35S::S11/15D-GFP* seedlings (Supplementary Fig. 15c), indicating that sim3|S11/15D is functional and responses to the stimulation of viral infection. Since sim3|S11/15D is located primarily in the cytoplasm[34], we suspected that sim3|S11/15D could be translocated into the nucleus during immune priming. Indeed, SA treatment remarkably increased the content of sim3|S11/15D-GFP in the nuclei of *35S::sim3|S11/15D-GFP* seedlings (Supplementary Fig. 15d).

To further confirm the function of sim3|S11/15D, we produced transgenic plants expressing a C-terminal Myc-tagged sim3|S11/15D under its native promoter in the *npr1-0* background (*npr1::sim3|S11/15D-Myc*). All homozygous *npr1::sim3|S11/15D-Myc* transgenic plants displayed a phenotype normal to that of WT plants under steady-state conditions (Supplementary Fig. 16a). Consistently, we found that *npr1::sim3|S11/15D-Myc-3* seedlings had a significantly smaller TuMV-infected leaf area and less viral genomic RNA that WT seedlings at 18 dpi (Supplementary Fig. 16b, c). These results indicate that the binding of NIb to SIM3 of NPR1 may also disrupt the phosphorylation of Ser11/Ser15. To confirm this hypothesis, we transiently expressed FLAG-4×Myc-SUMO3 and YFP-tagged NPR1 or its mutants in the epidermal cells

of *N. benthamiana* in the presence of RFP-tagged NIb, NIb$_{sim2}$, or GUS. The stimulation of agrobacteria was sufficient to translocate NPR1 into nuclei (Supplementary Fig. 4). At 2 dpi, nuclear NPR1 or its mutants were immunoprecipitated with GFP-Trap agarose and analyzed with a biotinylated Phos-tag. NPR1-YFP showed a strong phosphorylation signal, while the phosphorylation signal was dramatically reduced in S11/15A-YFP (Fig. 4f), suggesting that nuclear NPR1 was dominantly phosphorylated at Ser11/Ser15 in our experimental conditions. Interestingly, no phosphorylation signal was detected in sim3-YFP (Fig. 4f), confirming that phosphorylation of Ser11/Ser15 is dependent on SIM3-mediated sumoylation. As expected, the presence of NIb reduced about one third of the phosphorylation of NPR1, and deletion of SIM2 in NIb almost completely abolished this suppression (Fig. 4f). Together, these results indicate that NIb also suppresses the sumoylation-dependent phosphorylation at Ser11/Ser15.

## Targeting NPR1 SIM3 is a conserved ability of potyviral NIbs

Considering SIM2 and Lys409 are highly conserved among potyviral NIb proteins[32], we postulated that NIbs of most, if not all, potyviruses are able to interact with NPR1 and interrupt its function by targeting SIM3. The NIb genes of SMV, PVY, PRSV, bean common mosaic virus (BCMV), sugarcane mosaic virus (SCMV), and pepper veinal mottle virus (PVMV) were cloned, and their abilities to interact with Arabidopsis NPR1 or sim3 were analyzed. These potyviruses plus TuMV represent six major clades of the potyviral phylogeny[43]. Interestingly, we found that yeast cells transformed with NPR1 and either of the NIb proteins survived on the selective medium, while those cotransformed with sim3 failed (Fig. 5a, b and Supplementary Fig. 17a, b). Consistently, we found a bright YFP signal in the nuclei of *N. benthamiana* epidermal cells coexpressing either of the six YC-tagged NIb proteins and NPR1-YN but not sim3-YN in the BiFC assay (Fig. 5c, d). Western blotting confirmed that the failed interactions between sim3 and NIb proteins were not due to a lack of protein expression (Supplementary Fig. 17c, d). Moreover, we found that the presence of NIb proteins of SMV, PVY, BCMV, SCMV, PRSV, or PVMV could influence the luciferase activity from the SUMO3–NPR1 interaction to various degrees (Fig. 5e and Supplementary Fig. 17e). Together, these results suggest that binding of SIM3 of NPR1 to disrupt the NPR1–SUMO3 interaction is a highly conserved ability of potyviral NIb proteins.

## Soybean-encoded NPR1 is sumoylated by GmSUMO5

Although it is well established that SUMO3-mediated sumoylation regulates Arabidopsis NPR1 function[34], the conservation of this posttranslational modification of NPR1 is unknown since SUMO3 homologs have been found in only one clade of the *Brassicaceae* family[44]. Multiple sequence alignment and sumoylation predictions showed that SIM3 is highly conserved in NPR1 encoded by most plant species (Fig. 6a), indicating that NPR1 of other plants may also be regulated by sumoylation. To test this hypothesis, we cloned all six SUMO paralogs of soybean (GmSUMO1-GmSUMO6). Phylogenetic analysis showed that none of the soybean SUMO paralogs were phylogenetically related to Arabidopsis SUMO3 (Supplementary Fig. 18a). Y2H and BiFC assays showed that soybean NPR1 (GmNPR1) interacted exclusively with GmSUMO5 (Fig. 6b and Supplementary Fig. 18b–d). GmSUMO5 failed to interact with SIM3-mutated GmNPR1 (Gmsim3) in Y2H (Supplementary Fig. 18e, f), indicating that the interaction between GmNPR1 and GmSUMO5 is also dependent on SIM3. Moreover, we found that the NIb of SMV ($_{SMV}$NIb) also interacts with soybean NPR1 (GmNPR1) but not Gmsim3 (Fig. 6c and Supplementary Fig. 19). Co-IP assays showed that not only unmodified GmNPR1 but also the GmSUMO5-sumoylated form of GmNPR1 were pulled down by GmSUMO5 (Fig. 6d). To investigate whether $_{SMV}$NIb can interrupt the GmNPR1–GmSUMO5 interaction and sumoylation of GmNPR1 by

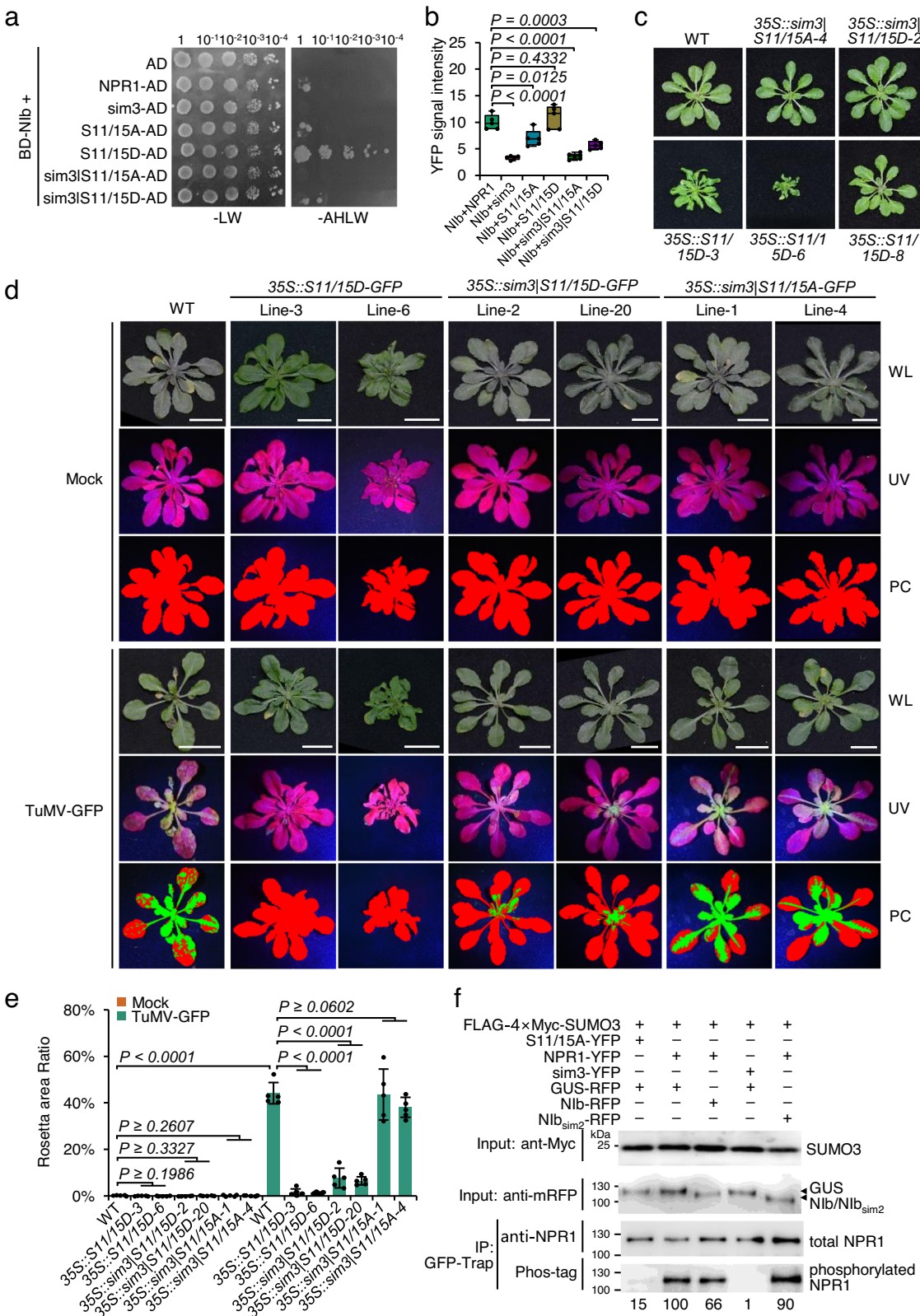

GmSUMO5, we performed Co-IP in the presence of $_{SMV}$NIb or GUS. At similar expression levels of GmSUMO5 and GmNPR1, the amount of nonmodified and posttranslationally modified GmNPR1 was greatly reduced in the presence of $_{SMV}$NIb compared with that of GUS (Fig. 6e). Together, these results suggest that soybean NPR1 also interacts with and is sumoylated by the noncanonical GmSUMO5 via SIM3 and that $_{SMV}$NIb disrupts this interaction.

## Discussion

The pivotal roles of SA signaling in PTI, ETI, and SAR have been well established; however, the contribution of SA signaling to compatible plant–virus interactions have remained ambiguous. In this study, we found that loss-of-function of *NPR1* significantly increased the susceptibility of Arabidopsis to TuMV and significantly reduced the expression of *PR1* after challenge with TuMV (Fig. 2f and

**Fig. 4 | NIb attenuates the phosphorylation of NPR1 at Ser11/Ser15. a** Evaluation of the interaction between NIb and NPR1 mutants by Y2H. The experiment was performed as shown in Fig. 1a except that the incubation time was reduced due to the fast growth rate of yeast cells transformed with BD-NIb and S11/15D-AD. **b** Box and whisker plot with individual data points showing the nuclear YFP intensity in *N. benthamiana* epidermal cells expressing NIb-YC and YN-tagged NPR1 or its mutants. The data are the mean of the nuclear signal intensity of 5 micrographs. The whisker indicates minimum and maximum, and the box indicates the 25th and 75th percentiles (edges of the box), and median (center line). The original figures are available in the Figshare repository under [https://doi.org/10.6084/m9.figshare.23243918]. **c** Phenotypes of 3-week-old WT, *35S::sim3|S11/15A-GFP*(*35S::sim3|S11/15A*), *35S::sim3|S11/1SD-GFP* (*35S::sim3|S11/1SD*), and *35S::S11/1SD-GFP* (*35S::S11/1SD*) seedlings under steady-state conditions. **d** Phenotypes of

mock or TuMV-infected WT and transgenic plants at 18 dpi. **e** Bar plot showing the ratio of virus-infected to total leaf area of WT and transgenic plants at 18 dpi ($n = 5$). **f** Immunoblot analysis of the phosphorylation of transiently expressed NPR1 or its mutants in the presence of GUS, NIb or NIb$_{sim2}$ by Phos-tag. Equal amounts of GFP-Trap agarose-enriched NPR1 or its mutants were analyzed by anti-NPR1 antibodies or treated with Phos-tag. Numbers represent relative Phos-tag intensities that are corrected according to the total protein amount. FLAG-4×Myc-SUMO3 and RFP-tagged GUS, NIb or NIb$_{sim2}$ were detected with anti-Myc and anti-mRFP antibodies, respectively. The experiment was independently repeated twice with similar results. Data are presented as mean values ± SD. Statistical analyses were performed using two-tailed Student's *t* test. Source data are provided as a Source Data file.

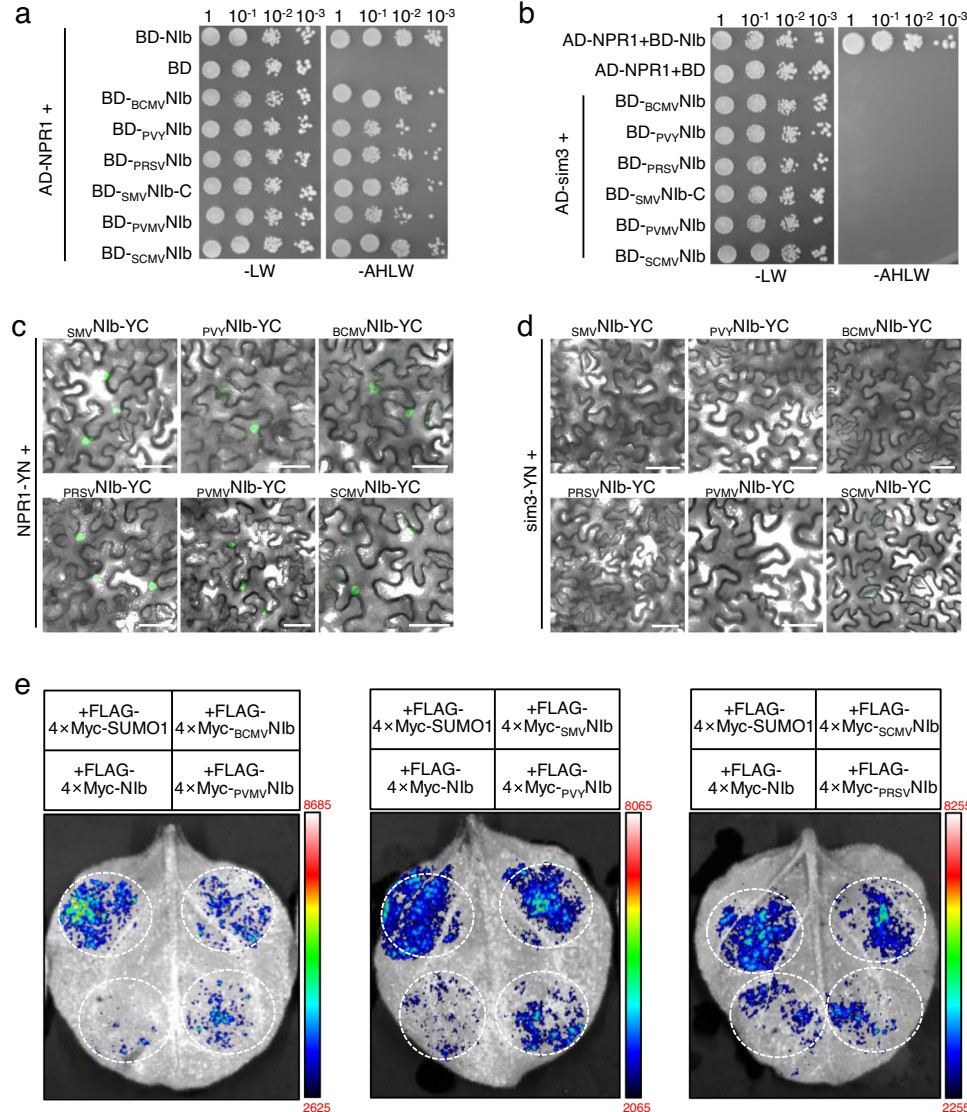

**Fig. 5 | Targeting SIM3 of NPR1 is a conserved function of NIbs from diverse potyviruses. a, b** Evaluation of interactions between NIb from different potyviruses and NPR1 (**a**) or sim3 (**b**) by Y2H. Due to the high-level of autoactivation of full-length NIb of SMV, only the C-terminal domain harboring SIM2 ($_{SMV}$NIb-C) was used. **c, d** Evaluation of the interaction between potyviral NIb proteins and NPR1 (**c**)

or sim3 (**d**) by BiFC. Micrographs were taken at 2 dpi. Scale bars = 50 μm. The experiment was independently repeated three times with similar results. **e** Influence of different potyviral NIb proteins on the luciferase activity of *N. benthamiana* leaf regions expressing cLUC-SUMO3 and NPR1-nLUC. All constructs were infiltrated at the same concentrations.

Supplementary Fig. 7a, b). These results suggest that the immune responses in the compatible Arabidopsis-TuMV pathsystem are largely dependent on NPR1; however, a minor NPR1-independent pathway also exists. How NPR1 is activated is currently unknown

since neither PTI nor ETI is believed to be activated in compatible plant–virus pathosystem. One possibility is that NPR1 is activated by viral-induced biotic stresses, e.g., endoplasmic reticulum (ER) stress, oxidative stress, and nutrition starvation stress[45,46]. Indeed,

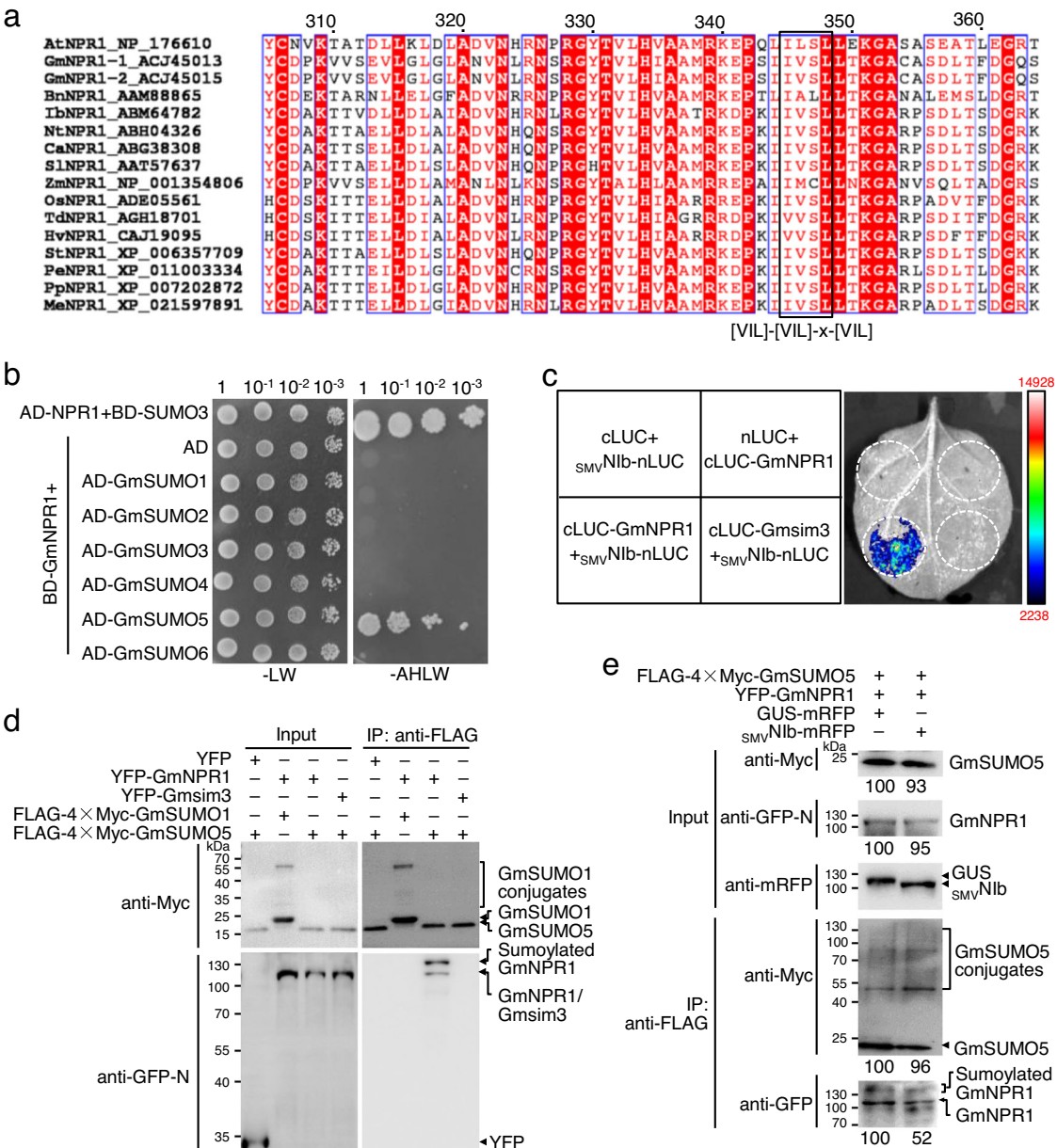

**Fig. 6 | Soybean NPR1 (GmNPR1) is sumoylated by GmSUMO5. a** Multiple alignment of partial amino acid sequences of NPR1 homologs of different plants. GenBank accession numbers are indicated at the front, and SIM3 is highlighted by a black box with a conserved aa pattern on the bottom. **b** Evaluation of the interaction between GmNPR1 and GmSUMO homologs by Y2H assay. **c** Evaluation of the interaction between SMV NIb ($_{SMV}$NIb) and GmNPR1 and Gmsim3 by split-luciferase assay. **d** Analysis of the sumoylation of GmNPR1 by Co-IP assay. Proteins were expressed in *N. benthamiana* leaves by agroinfiltration, analyzed by western blotting (left panel) or immunoprecipitated with anti-FLAG M2 affinity gel and then detected with anti-Myc and anti-GFP-N antibodies (right panel) at 2 dpi. Sumoylated GmNPR1 is indicated. To avoid overexposure, samples that were detected with anti-Myc antibodies were diluted 20 times for SDS–PAGE. The experiment was independently repeated twice with similar results. **e** Western blot analysis of the influence of $_{SMV}$NIb on GmNPR1 sumoylation. Proteins were expressed in *N. benthamiana* leaves and analyzed at 2 dpi by Western blotting with anti-Myc, anti-GFP-N, and anti-mRFP polyclonal antibodies (left panel). GmSUMO5 was immunoprecipitated by anti-FLAG M2 affinity gel and then analyzed by anti-Myc or anti-GFP antibodies. Nonmodified and posttranslationally modified GmNPR1 are indicated by an arrow and square bracket, respectively. Similar results were observed in three independent experiments.

the unfolded protein response (UPR) during ER stress can cause the translocation of NPR1 to the nucleus[47]. We are trying to confirm this possibility at present.

In the nucleus, NPR1 is subjected to several posttranslational modifications, e.g., phosphorylation, sumoylation, and ubiquitination[15,34]. It is believed that SUMO3-mediated sumoylation and the phosphorylation of Ser11/Ser15 form a signal amplification cycle for robust immune reprogramming[34]. Interestingly, our results revealed that the overexpression of a phosphorylation-mimic NPR1 mutant (S11/15D) caused an autoimmunity phenotype (Fig. 4c).

Moreover, the protein levels of S11/15D in those transgenic plants were positively correlated with phenotype severity, *PR1* expression, and TuMV resistance (Fig. 4). The autoimmunity-inducing capacity of S11/15D has not been noticed in previous studies, possibly due to low expression levels in those transgenic lines[15,34]. Nevertheless, our data strongly suggest that phosphorylation at Ser11/Ser15 is essential for NPR1 to vigorously reprogram the transcription profile during immune responses. Our results also showed that *35S::sim3|S11/15A-GFP* plants were as susceptible to TuMV as *35S::sim3-GFP* plants, while *35S::sim3|S11/15D-GFP* plants displayed significantly enhanced resistance (Fig. 4).

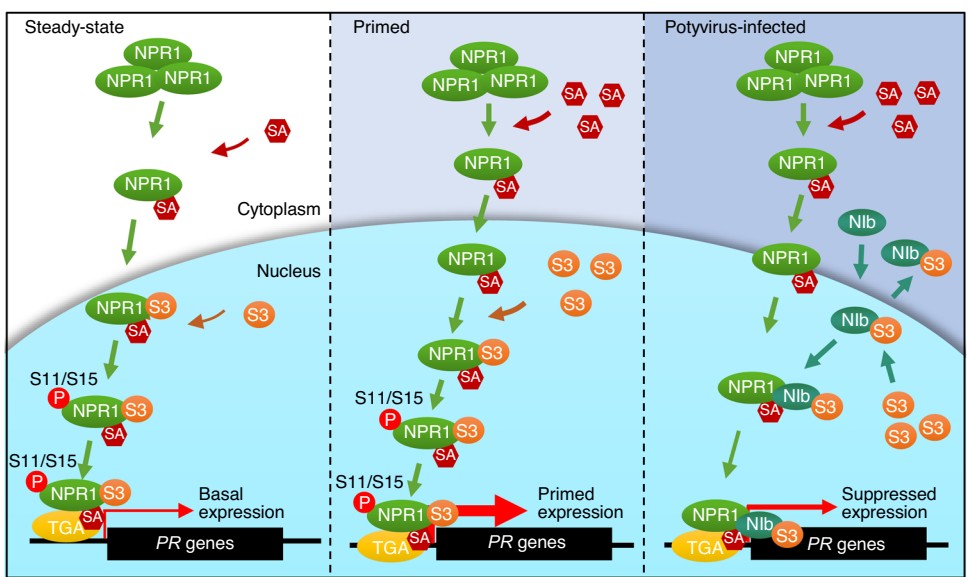

**Fig. 7 | Schematic model of how NIb inhibits NPR1-mediated immune responses.** Under steady-state conditions (left panel), monomeric NPR1 that is released from oligomers binds to SA and is imported into the nucleus, where it is sumoylated by SUMO3 via SIM3, is phosphorylated by a thus far unknown kinase at residues Ser11/Ser15, and then associates with TGA transcription factors to maintain basal expression of defense-related genes. Under immunity-primed conditions (middle panel), an increased level of SA triggers rapid and immense release of NPR1 monomers into the nucleus, where they active massive expression of defense-related genes, including *SUMO3*[33]. Under TuMV infection conditions (right panel), newly synthesized NIb is rapidly imported into the nucleus, where it binds to SIM3 of NPR1, which prevents NPR1–SUMO3 interaction and sumoylation of NPR1 by SUMO3. Meanwhile, sumoylation of NIb by SUMO3 increases the affinity of NIb for NPR1, which forms a positive feedback effect. The binding of NIb also inhibits downstream phosphorylation at Ser11/Ser15, which ultimately results in the attenuation or even shutdown of NPR1-mediated immunity. Sumoylated NIb is also exported by XPO1 to promote viral replication[59]. P, phosphate, S3, SUMO3.

An in vitro phosphorylation assay showed that phosphorylation of NPR1 at Ser11/Ser15 is completely dependent on SUMO3-mediated sumoylation (Fig. 4f). The paralog of SUMO3 is present in plants of one clade in the *Brassicaceae* family[44], which raises a key question: Do NPR1 paralogs in plants have no SUMO3-related paralogs that are also subjected to sumoylation? Our results showed that soybean-encoded NPR1 is also sumoylated by the noncanonical gene GmSUMO5, which is not phylogenetically related to SUMO3 (Fig. 6). Moreover, the sumoylation of GmNPR1 by GmSUMO5 also occurs via SIM3 and SMV-encoded NIb can disrupt this interaction. These results thus confirm the importance and conservation of sumoylation in regulating NPR1 activity for proper defense gene induction.

The most interesting finding of this study is the discovery that TuMV-encoded NIb interacts with and interrupts NPR1 function. Given the important role of the SA signaling cassette in plant defense, it is not surprising that some pathogens may disrupt SA signaling. For instance, *Pseudomonas syringae* produces a phytotoxin coronatine (COR), a structural mimic of jasmonates (JAs), to stimulate the transcription of JA-dependent genes and consequently attenuate SA-dependent genes expression[48]. AvrPtoB of *Pseudomonas syringae*, RxLR48 of *Phytophthora capsica*, and PUCCINIA NPR1 INTERACTOR (PNPi) of *Puccinia striiformis* have been found to target NPR1[49–51]. RxLR48 promotes nuclear accumulation of NPR1 and inhibits NPR1 degradation; AvrPtoB mediates the degradation of NPR1 by the host 26S proteasome through its E3 ligase activity in the presence of SA; and PNPi interacts with the C-terminal domain of NPR1 and disrupts the NPR1–TGA2 interaction[49–51]. Our results show that NIb interacts with NPR1 exclusively in the nucleus and that the interaction is not affected by SA (Fig. 1b and Supplementary Fig. 2b), suggesting that NIb targets a step that occurs after NPR1 has entered the nucleus. Moreover, we found that NIb directly targets SIM3 of NPR1 to disrupt the NPR1–SUMO3 interaction and suppress SUMO3-mediated sumoylation (Fig. 3), which also disrupts the phosphorylation of the first IκB-like phosphodegron (Fig. 4). Importantly, our results illustrate that targeting SIM3 of NPR1 is a conserved capacity of potyviral NIbs (Fig. 5). Therefore, our

discovery represents a conserved strategy to suppress NPR1-mediated immunity by potyviral NIb proteins (Fig. 7). Given that transgenic plants expressing sim3|S11/15D have no obvious alternations in plant growth and basal defense-related gene expression but have significant increased resistance to TuMV, it is possible to improve the resistance of crops such as soybean and potato to potyviruses by editing their NPR1 genes.

It is well known that SA treatment provides broad-spectrum defense against many viruses, including both plant RNA and DNA viruses[20]. However, no obvious differences in the timing and severity of symptoms are observed between *npr1* knock-out plants and wild-type plants upon infection with a few viruses, e.g., cabbage leaf curl virus (CaLCuV; genus *Begomovirus*), cucumber mosaic virus (CMV; genus *Cucumovirus*), and oilseed rape mosaic virus (ORMV; genus *Tobamovirus*)[23,52]. These results indicate that NPR1 is at the hub of the wrestling match between plants and viruses and that the function of NPR1 may have already been completely or almost completely suppressed by some of these viruses. It will be interesting to investigate how these viruses subvert NPR1-mediated defenses in the compatible interaction.

Due to their small genome size and low encoding capacity, viruses have evolved various strategies for their survival. An important one is to make viral protein(s) multifunctional. NIb localizes both in the nucleus and cytoplasm[32]. It is believed that cytoplasmic NIb is mainly involved in viral replication, as it is the only viral protein containing conserved motifs of RdRp. Indeed, several host proteins involved in RNA metabolism, e.g., EUKARYOTIC ELONGATION FACTOR 1A (eEF1A), POLY(A)-BINDING PROTEIN (PABP), HEAT-SHOCK PROTEIN 70-3 (Hsp70-3), and WHEAT LIGHT-INDUCED PROTEIN (TaLIP), have been found to be coopted by NIb for robust infection[53–56]. Moreover, cytoplasmic NIb is also targeted by the host surveillance system; e.g., NIb is targeted by beclin1 for degradation and is recognized by the broad-spectrum resistance gene *Pvr4* in *Capsicum annuum* cv. CM334[57,58]. Previously, we discovered that NIb exhibits a key function of in the nucleus by interacting with sumoylation machinery to subvert

host defenses[31,32]. Results of this study demonstrate that NIb directly targets SIM3 of NPR1 to compete for SUMO3 and suppress host defense. Sumoylated NIb may be exported from the nucleus by EXPORTIN 1 (XPO1) for infection[59]. In this study, we further showed that NIb likely interacts with all known SUMO3 substrates that are involved in varied pathways and biological processes. NIb may also influence their functions and the pathways they are involved, which further implies the versatile functions of NIb. Attachment of SUMO typically marks a protein for recognition by a SIM-containing protein[60]. Bioinformatic analyses have revealed that more than 40% of Arabidopsis proteins contain a sumoylation site (ΨKXE/D, where Ψ is a hydrophobic amino acid and X represents any amino acids) and/or a SIM motif (I/VDL/T)[61–63]. How NIb disarranges the plant SUMO interactome is another engaging subject for exploration in the future.

## Methods

### Plant materials, virus inoculation, and sampling

Plants were grown in pots at 23°C in a growth chamber under a 16/8-h photoperiod with 50% humidity. *npr1-0*, *npr1-1*, *npr1-6*, and *35S:npr1$^{sim3}$* have previously been described[16,34,39,64]. Arabidopsis was transformed using the flower dip method[65]. Progeny seeds were screened by directly spraying 20 mg/L Basta aqueous solutions or on 1/2 Murashige and Skoog (MS) medium plates supplied with hygromycin B (50 ng/mL). Agroinfiltration and sap inoculation were performed as described earlier with seedlings of the same age and similar sizes[32]. To minimize artificial variations, the whole aerial part of seedlings was harvested and ground in liquid nitrogen, and equal amounts of plant material were used for subsequent analyses, e.g., total RNA extraction, immunoblotting, nucleocytoplasmic partitioning, and ELISA.

### Plasmid construction

Full-length or partial coding sequences of *NPR1* (At1g64280), *TCP3* (At1g53230), *TCP4* (At3g15030), *TCP23* (At1g35560), *BRM* (At2g46020), *SPL4/FTM6* (At1g53160), *CRF6* (AT3G61630), *TGA3* (At1g22070), *ERF5* (At5g47230), *FBH3* (At1g51140), mature form of *SUMO3* (At5g55170) and *NIb* of TuMV (NC_002509), SMV (MN623290), PVY (MH933741), BCMV (KP903372), PVMV (MN082715), SCMV (KR108213), and PRSV (HQ424465) were amplified using the primers listed in Supplementary Table 1 with Phanta superfidelity DNA polymerase (Vayzme, Nanjing, China) and inserted into the pDONR207 vector with a ClonExpress II One Step Cloning Kit (Vayzme). NPR1 and NIb point mutants were generated by overlapping PCR using the primers listed in Supplementary Table 1. The gateway-compatible pEarleyGate-101 and pEarleyGate-104 plant binary expression vectors were used to produce C- or N-terminal YFP-tagged constructs, pEarleyGate-104 was used to construct C-terminal GFP-tagged constructs[66], and pBA-FLAG-4×Myc-DC was used to produce the N-terminal FLAG-4×Myc-tagged construct[67]. The gateway-compatible vectors pGBKT7-DEST and pGADT7-DEST were used to produce plasmids for the Y2H assay[68], pEarleyGate201-YN, pEarleyGate202-YC, YC-pEarleyGate100, or YN-pEarleyGat100 were used to generate constructs for the BiFC assay[68], and the Gateway compatible vectors pCAMBIA-NLuc-GW and pCAMBIA-CLuc-GW, which were modified from pCAMBIA1300-NLuc and pCAMBIA1300-CLuc, respectively[69], were used for the split-luciferase assay. The vectors pET-32a (+) (Merck Millipore, Beijing, China) and pGEX-4T-1 (Cytiva, Marlborough, MA, USA) were used to express N-terminal Trx-6×His and GST-tagged recombinant proteins in *Escherichia coli*, respectively. All plasmids were verified by Sanger sequencing.

### Yeast two-hybrid assay

Y2H assays were performed using the yeast strain Golden (Clontech, Beijing, China) as described previously[32]. All yeast selective media were purchased from Coolaber Technology (Beijing) Co., Ltd.

### Split-luciferase assay

To determine the luciferase activity, *N. benthamiana* leaves were agroinfiltrated with agrobacteria harboring proper plasmids. The infiltrated leaves were integrally detached from the plant at 60 hpi and soaked in 10 μM D-luciferin (Sangon, Shanghai, China) for ~20 min. The leaves were then visualized with a Tanon 5200CE Chemiluminescent Imaging System (Tanon Science & Technology Co., Ltd).

### Total protein extraction

For total plant protein, ~0.1 g of fine powder of the aerial parts of the plant or agroinfiltrated leaf tissues was resuspended in 100 μL of 1× sodium dodecyl sulfate–polyacrylamide gel electrophoresis (SDS–PAGE) loading buffer (2% SDS, 10% glycerol, 100 mM DTT, 0.005% bromophenol blue and 50 mM Tris HCl [pH 6.8]). After boiling at 95 °C for 5 min, the crude lysate was centrifuged at 12,000×*g* for 10 min at 4 °C, and the supernatant was stored at 80 °C until use or directly used for SDS–PAGE. For total yeast protein, a total volume of 5 mL of yeast overnight culture was used for total protein extraction using the yeast total protein extraction kit (Coolaber) according to the supplied protocol.

### Coimmunoprecipitation

Co-IP was performed as described previously with few modifications[32]. In brief, ~3 g leaf tissues were ground into fine powder in liquid nitrogen, mixed with 15 mL IP buffer I (50 mM Tris-HCl, pH 7.5, 150 mM NaCl, 0.1% Triton X-100, 0.1% NP-40, 5 mM EDTA, 2 mM DL-dithiothreitol [DTT], 5% glycerin, 1 mM phenylmethylsulfonyl fluoride [PMSF], and 1/2 tablet of Complete Protease Inhibitor [Roche, Shanghai, China]), and placed on ice for 15 min. After being filtered through two layers of Miracloth (Merch Millipore), the solution was sonicated 10 s at 10% power level eight times with a 1 min interval. The lysate was then centrifuged at 6000×*g* for 25 min at 4 °C. The resulting supernatant was added to 30 μL of Chromotek GFP-Trap (Proteintech, Wuhan, Hubei, China) or anti-FLAG M2 affinity gel (Sigma–Aldrich, Shanghai, China) and incubated for 3 h at 4 °C with gentle rotation. Agarose beads were collected by centrifugation, washed at least five times with IP Buffer and resuspended in 50 μL 1× SDS–PAGE loading buffer. The solution was frozen at −80 °C until use or directly used for SDS-PAGE.

### SDS–PAGE and immunoblotting

SDS–PAGE was performed using 10 or 12% homemade polyacrylamide gels. After SDS-PAGE, proteins were transferred to polyvinylidene fluoride (PVDF) membranes with a Trans-Blot Turbo Transfer System (Bio-Rad). The PVDF membranes were rinsed briefly in TBST buffer and then blocked with 5% nonfat dry milk (in TBST buffer) for 1 h at room temperature. The membranes were incubated with proper antibodies at the desired dilution at room temperature for 1 h or overnight at 4 °C. After washing six times with TBST, the membranes were incubated with the appropriate secondary antibodies (Sigma–Aldrich) for 1 h at room temperature. After washing six times with TBST, the membranes were visualized using the Immobilon Western Chemiluminescent HRP Substrate (Merch Millipore) following the manufacturer's instructions with a Tanon 5200CE Chemiluminescent Imaging System. In each experiment, a parallel gel was stained with Coomassie Brilliant Blue as a loading control.

The following antibodies were used: rabbit anti-GFP N-terminal (Sigma–Aldrich) at a 1:5000 dilution, mouse monoclonal anti-GFP antibodies (Roche) at a 1:1000 dilution, rabbit anti-SUMO3 (Abcam, Shanghai, China) at a 1:2000 dilution, polyclonal anti-Myc (Abcam) at a 1:2000 dilution, mouse monoclonal anti-GAL4 AD [14-7E10G10] antibody (Abcam) at a 1:2000 dilution, Rabbit polyclonal to Histone H3 (Abcam) at a 1:5000 dilution, anti-Luciferase polyclonal antibody (Proteintech) at a 1:10,000 dilution, mouse monoclonal anti-RFP (Proteintech) at a 1:5000

dilution, HRP-conjugated anti-rabbit IgG secondary antibodies (Sigma-Aldrich) at a 1:10,000 dilution, and HRP-conjugated anti-rabbit IgG secondary antibodies (Sigma-Aldrich) at a 1:10,000 dilution. Rabbit polyclonal antibodies against TuMV CP (anti-CP), TuMV NIb (Anti-NIb), and Arabidopsis NPR1 (Anti-NPR1) were produced by ABclonal Biotechnology co., Ltd (Wuhan, China) using GST-tagged recombinant proteins of full-length CP, amino acids 165-488 of NIb, and amino acids 1-465 of NPR1, respectively. Both of these antibodies were used at a 1:1000 dilution. Validation data of anti-CP, anti-NIb, and anti-NPR1 have been provided as a Source Data file.

### Detection of phosphorylated protein
The phosphorylated protein was detected on a PVDF membrane with the biotinylated Phos-tag (l,3-bis[bis(pyridine-2-ylmethyl) amino] propan-2-olato dizinc (II) complex (BTL-105) from APExBIO Technology LLC (Houston, Texas, USA) and streptavidin-HRP (Thermo Fisher Scientific, Shanghai, China). In brief, PVDF membranes that were blocked with 5% nonfat dry milk (in TBST buffer) for 1 h at room temperature were incubated in 10 mL of Phos-tag biotin solution (TBST supplemented with 6 μM $Zn(NO_3)_2$, 3.5 μM Phos-tag Biotin, and 0.15 μL/mL streptavidin-HRP) for 1 h at room temperature. After washing five times with TBST for 5 min each time at room temperature, the membrane was soaked in 500 μL of Immobilon Western chemiluminescent HRP substrate solution and visualized with a Tanon 5200CE Chemiluminescent Imaging System.

### Nucleocytoplasmic partitioning
Approximately 0.1 g fine powder of the aerial parts of the plant was resuspended in 200 μL of extract buffer (0.4 M sucrose, 10 mM Tris-HCl pH 8.0, 10 mM $MgCl_2$, 5 mM β-mercaptoethanol, 0.1 mM PMSF, and 1/2 tablet of Complete Protease Inhibitor). After being filtered through two layers of Miracloth, the lysate was centrifuged at 4000×$g$ for 20 min at 4 °C. The resulting supernatant was added to 70 μL of 4× SDS–PAGE loading buffer, mixed well, boiled for 5 min, and then directly used for SDS-PAGE. The resulting pellets were mixed with 200 μL of nuclear extraction buffer (50 mM Tris-HCl pH 7.5, 150 Mm NaCl, 1% NP40, 2 mM EDTA, and 1/2 tablet of Complete Protease Inhibitor) and 70 μL 4× SDS–PAGE loading buffer. After vortex mixing, the mixture was boiled for 5 min and then directly used for SDS-PAGE. Equal amounts of supernatant and pellets were used for immunoblotting with exactly the same treatments and conditions.

### ELISA
ELISA was performed using polyclonal antiserum to the TuMV CP. Goat anti-rabbit immunoglobulin G conjugated with alkaline phosphatase (Agdia, Elkhart, IN, USA) was used as a secondary antibody, and p-nitrophenyl phosphate (Agdia) was used as the substrate for color development.

### Confocal microscopy
Agrobacteria harboring the proper plasmid were infiltrated into N. benthamiana leaves at the indicated OD values. The fluorescence was visualized with a TCS SP8 LIGHTNING Confocal Microscope (Leica, Wetzlar, Germany) as described previously[70]. Sequential mode was used when multiple fluorescent proteins were recorded.

### Prokaryotic protein expression, purification, and GST pulldown
Recombinant proteins were expressed in E. coli BL21 (DE3) cells with 0.5 mM isoprophyl-β-D-thiogalactoside (IPTG) at 16 °C overnight. Trx-His and GST-tagged proteins were purified with Ni–nitrilotriacetic acid (Ni-NTA) agarose (Thermo Fisher Scientific) and glutathione agarose (Thermo Fisher Scientific), respectively. For the GST pulldown assay, equal amounts of recombinant GST and GST-NIb were incubated with

Trx-His or Trx-His-NPR1 and GST beads at 4 °C for 2 h. After washing with wash buffer (100 mM NaCl, 20 mM Tris-HCl [pH 7.5], 0.05% NP-40) six times, the GST beads were boiled for 5 min in 2× SDS–PAGE loading buffer. Equal amounts of supernatants were loaded onto a 10% acrylamide gel, separated by electrophoresis, and detected with antibodies against GST or 6×Histidine.

### Quantitative analysis
Relative quantification of protein bands in Western blotting was performed by densitometric analysis using GelAnalyzer 19.1 software (http://www.gelanalyzer.com/) according to the instructions provided.

For quantitative analyses of confocal microscopic images, the fluorescence of target protein(s) was excited and recorded at minimized intensity (e.g., laser intensity 3% and PMT gain 500) to avoid possible overexposure, while the signal of marker protein was recorded at optimized intensity (e.g., laser intensity 10% and PMT gain 800). Fluorescence quantitative analyses were performed using Fiji, a version of ImageJ2 for scientific image analysis[71]. In brief, nuclei were determined by the fluorescence of the nuclear marker with a size above 15 pixels and circularity between 0.2 and 1.0 and overlaid on the image of BiFC signals for analysis. The fluorescence of target protein(s) in the nuclei was then determined using the particle analysis function in Fiji with default parameters and the average value of the nuclear fluorescence of the micrograph was calculated.

Images of Arabidopsis plants were taken using a Nikon DH-SLR D5200 camera. Images under UV light were split into red, green, and blue channels using Fiji[71]. The total leaf area was determined by the red channel pixels, while the virus-infected area was assessed by the green channel pixels using the Analyze Particles function with size 100-infinity and circularity 0.0–1.0. A pseudocolor image was also produced with the ROI manager function.

### Reverse transcription and quantitative PCR
Total RNA was isolated using the Eastep® Super Total RNA Extraction Kit (Promega, Beijing, China) according to the manufacturer's protocol. Complementary DNA (cDNA) was synthesized by Oligo $(dT)_{20}$ or random hexamers using a HiScript III 1st Strand cDNA Synthesis Kit with gDNA wiper (Vazyme) as instructed. RT-qPCR was performed using a Roche LightCycler 96 System in a 20 μL volume system containing 4 μL of 10-fold-diluted cDNA, 5 μM of each primer, and 1× AceQ® Universal SYBR qPCR Master Mix (Vazyme). The genomic RNA of TuMV was determined by amplification of a 257 bp fragment of the TuMV CP gene. The Arabidopsis PR1 (AT2G14610), UBQ5 (AT3G62250), GAPDH (AT3G26650), and ACTIN II (AT3G18780) were amplified with primers listed in Supplementary Table 1.

### Phylogenetic analyses
The phylogenetic tree was constructed using MEGA 11 software with the neighbor-joining method[72]. The confidence of the phylogenetic tree was tested by bootstrap method with 1000 replicates. Multiple alignment was performed using the ClustalW version 2.0[73] and rendered using the ESPript3[74].

### Statistical analysis
All statistical analyses were performed by two-tailed Student's t test with Office Excel 2016 or GraphPad Prism 8.0.2. All data are represented as the mean ± SD. Source data are provided as a Source Data file.

### Reporting summary
Further information on research design is available in the Nature Portfolio Reporting Summary linked to this article.

## Data availability

All data are available within the Article and Supplementary Files. All constructs and transgenic plants are available upon request. The small RNA-seq data generated in this study have been deposited in the GenBank database under accession codon: PRJNA877833. The raw data of Figs. 3a and 4b are available in the Figshare repository under https://doi.org/10.6084/m9.figshare.23243918. Source data are provided with this paper.

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

## Acknowledgements

We are grateful to Dr. Xinnian Dong from the Duke University for providing the *npr1-1* and *35S:npr1^sim3*, Xiu-fang Xin from the Institute of Plant Physiology and Ecology of the Chinese Academy of Sciences for sharing *npr1-6*, Dr. Hongguang Cui from Hainan University for providing PVMV-encoded NIb, Dr. Wentao Shen from Chinese Academy of Tropical Agricultural Sciences for providing PRSV-encoded NIb, and Dr. Jinsheng Xu from Fujian Agriculture and Forestry University for providing SCMV-encoded NIb. This work is financial supported by the National Natural Science Foundation of China (32022071 to X.C.), the Natural Science Foundation of Heilongjiang Province (ZD2018002 to X.C.; LH2019C027 to X.W.), the Program for the Scientific Activities of Selected Returned Overseas Professionals in Heilongjiang Province (2018QD0002 to X.C.), and the "Young Talents" Project of Northeast Agricultural University (18XG04 to X.W.). The work in A. Wang laboratory was supported in part by the Agriculture and Agri-Food Canada (AAFC) and the Natural Sciences and Engineering Research Council of Canada (NSERC).

## Author contributions

J.L., X.W., and X.C. conceived and designed research. J.L., X.W., Y.F., Y.Liu, Y.Li, and E.O.B. performed experiments and/or analyzed data; R.X. and Y.L. contributed materials and analytic tools; and X.W., Z.Q.F., A.W., and X.C. wrote and corrected the paper.

## Competing interests

The authors declare no competing interests.
