## [Peer Review File · Nature Communications]

Reviewer comments, first round –

Reviewer #1 (Remarks to the Author):

SUMMARY

This manuscript shows that to promote virus infection potyviruses deploy NIB to suppress NPR1-mediated resistance. Potyviral NIB disrupts sumoylation by binding to the SUMO-interacting motif 3 (SIM3) of NPR1. This binding prevents interaction of NPR1 with SUMO3 and sumoylation, and phosphorylation of NPR1. Sumoylation of NIB by SUMO3 is not essential but can intensify NIB-NPR1 interaction. This is a novel mechanism of use by viruses to overcome antiviral immunity.

MAIN CONTRIBUTIONS

1. *npr1* mutants are hypersusceptible to TuMV.
1. NIB encoded by TuMV interacts with NPR1.
2. NIB binds the SUMO-interacting motif 3 (SIM3) in the ankyrin repeat domain of NPR1.
3. NIB interferes with NPR1-SUMO3 interaction and suppresses the sumoylation-dependent phosphorylation of NPR1.
4. Binds the SUMO-interacting motif 3 (SIM3) in NPR1 is a property of NIB encoded by several potyviruses.
5. Observations are not limited to Arabidopsis. Soybean-encoded NPR1 is sumoylated by GmSUMO5.

POINTS THAT NEED TO BE CLARIFIED

1. The model in figure 7 needs to be simplified and updated to accurately reflect the data. In its current form, the panel corresponding to potyvirus-infected is contradictory to PR1 expression levels in Fig. 1e and Supplemental Figure 3.

Reviewer #2 (Remarks to the Author):

In this manuscript, the authors attempt to use a series of results to demonstrate that the Turnip mosaic potyvirus (TuMV) uses its RNA-dependent RNA polymerase NIB to interfere with NPR1 sumoylation, resulting in suppression of NPR1 phosphorylation at S11/S15, which in turn weakens the host immune responses. Based on these results, the authors propose that TuMV and other potyviruses deploy NIB to suppress NPR1-mediated resistance by targeting its sumoylation and consider this mechanism “a novel molecular arms race”.

The NIB-SUMO3-NPR1 triangle is certainly interesting. However, the manuscript focused on one side of the coin and might not reveal the true relationship among the three players.

Major concerns

1. NIB is the enzyme that replicates the viral genome. After translation, NIB is transported into the nucleus where it is sumoylated by SUMO3 and subsequently shuttled back to the cytoplasm to replicate the viral genome. Nuclear NPR1 not only competes for SUMO3 but also directly interacts with NIB. It is thus possible that NPR1 also suppresses NIB activity or keeps NIB in the nucleus. This hypothesis is supported by Figures 2d to 2h. Although PR1 was similarly induced in the WT, NPR1-GFP, *sim3*-GFP, and *npr1-sim3* plants, NPR1-GFP showed significantly enhanced resistance, whereas *npr1-sim3* appeared to be more susceptible than the wild type (Figures 2g and 2h). If PR1 is used as the marker of immune responses, it can be concluded that it is the NPR1-GFP protein, not immune responses, that suppresses the viral propagation. Is it possible to test if NPR1 protein could suppress NIB activity? The T-DNA insertion line SAIL_708_F09 that accumulates a truncated NPR1 protein containing the ANK repeats (Ding et al., 2020 Front Plant Sci) would be a good material to uncouple NPR1-mediated immune responses and the ANK domain-mediated

interaction with NIB.

The results of S11/15D in Figure 4 further support the function of NIB-NPR1 interaction in suppressing NIB activity. The authors may want to show TuMV-induced PR1 expression in the lines used in Figure 4. The S11/15D protein has a much stronger interaction with NIB and the S11/15D plants were not infected by the virus. The sim3/S11/15D plants are also resistant due to the new interaction created by the S11/15D mutation.

It is possible that the NIB-NPR1 interaction suppresses the activity of both proteins. The authors showed the effect on NPR1; but the effect on NIB activity may play a more important role in the plant-virus interaction.

2. Both NIB and NPR1 are conserved components in their own systems. It would be difficult to imagine that the virus and the plant use such important proteins for an arms race. The NIB-NPR1 interaction may be fortuitously formed and may not be a result of evolution under certain selections.

3. The results in Figures 2g and 2i are quite different based on the p values, but it is stated that "a similar trend of accumulation of viral RNAs in those transgenic plants". GFP silencing cannot be excluded.

Minor comments

1. Page 10 line 237, it may be more likely the SIM2 site is involved in the interaction.

2. Page 12 line 291, the fact that PR1 expression levels are correlated with S11/15D protein levels does not indicate that the phosphorylation is essential.

3. Page 13 line 309, "these results indicate that binding of NIB to SIM3 of NPR1 not only inhibit NPR1-SUMO3 interaction, but also disrupts the phosphorylation of S11/15". Results in Figures 4a to 4f really indicate that S11/15 phosphorylation enhances the NIB-NPR1 interaction and resistance to the virus. Figure 4g shows reduced phosphorylation.

4. Figure 2b, bright-field images would be helpful. Figure 3a magnification should be increased.

5. Figure 3g, anti-SUMO3 is not in the figure.

6. Page 38, Supplemental fig. 12 legend, d analysis of S11/15D transcript, but in the figure it is PR1.

Reviewer #3 (Remarks to the Author):

In this paper, Liu et al further analyse the role of NPR1 sumoylation on arabidopsis susceptibility to TuMV. They show that NPR1 is involved in slightly repressing TuMV progression in arabidopsis but that this repression is slowed down by the action of the TuMV NIB that inhibits NPR1's sumoylation.

It is of interest, but I am not sure I completely grasp how significant this arms race means for the interaction between plant and virus. Besides, I am some technical points that might need clearing. Overall, the paper present an impressive set of experiments but I wonder if the number of repeats or the effectives are sufficient to conclude at each step.

1- I did not find the starting point very clear (or it is very lucky). (page 5-6 lines 119-122. Was NPR1 the only candidate picked ? This result is highlighted by the authors as the most interesting of the study (page 16, line 402), yet from the begining of the results section it looks like only NPR1 was tested as a potential interactant for NIB ? Is that so ? Wasn't this supported by other data (Y2H screening of a library .. ?)

2- A lot of this paper relies on quite slight differences in virus accumulation. Total accumulation of

virus is carried out at 14 dpi, which is quite early. I wonder what happens at longer time after infection, and if the overall increased susceptibility is meaningful over time. Also, I could not find how was tissue sampling carried out for viral and CP quantification (Figure 1 h-i-j). Because at that stage the infection is very unequal in the plant (as seen by GFP imaging in figure 1-f), the sampling can have a very profound effect on the quantification, knowing that the number of samples (6) is quite low. Was the whole plant ground or only selected leaves ?

3- Page 9 line 219 and following, figure 2i, a similar trend was observed with 6K2 mCherry : I don't see that on the figure, to me (and statistically), all samples looks like they accumulate the same level of virus.

4- Figure 3-e, is it possible to conclude and make statistics on n=5 nuclei (is the the right n?)? Does it mean only one sample (leaf) was tested per condition ?

5- Page 11 line 260. « Results show that the sumoylated NPR1 was reduced by about 60% at the presence of Nib. » I am not a specialist but is Fig 3g sufficient to conclude ? How many repeats were done ? I would have the same question for the split-luciferase (Fig3F and 3H)

6- Targeting the SIM3 of NPR1... Which NPR1 is used for interaction studies ? Is it the arabidopsis one. It might be of interest to show how homologs Arabidopsis NPR1 and the one from species that are natural host for the tested viruses (unless they all infect arabidopsis ?) actually, it is in figure 6a ?

7- Discussion, page 15, lines 366-367. I am not sure whether it is that striking. PR1 induction on TuMV infection is very much reduced in a npr1 KO, but this is accompanied by a very modest increase in virus load, and quite early on. Therefore, i think it is interesting, but I am not sure it is very significant.

Minor points :

Line 187, please rephrase, missing words ?

To me, all Y2H experiments should be supported by western blot to ensure that a lack of interaction is not caused by the truncated/mutated proteins being unstable, this is especially true for experiments aiming at analysing quantitative interactions (Fig 3C, 4D) but I leave it at the editor's discretion

Page 18 line 444, I am not sure what to make of reference 58, where Nib is simply described as an avirulence factor

Page 18 line 447, word missing (be ?)

Point-to-point response to the reviewers' comments

REVIEWER COMMENTS

Reviewer #1 (Remarks to the Author):

SUMMARY

This manuscript shows that to promote virus infection potyviruses deploy Nib to suppress NPR1-mediated resistance. Potyviral Nib disrupts sumoylation by binding to the SUMO-interacting motif 3 (SIM3) of NPR1. This binding prevent interaction of NPR1 with SUMO3 and sumoylation, and phosphorylation of NPR1. Sumoylation of Nib by SUMO3 is not essential but can intensify Nib-NPR1 interaction. This is a novel mechanism of use by viruses to overcome antiviral immunity.

MAIN CONTRIBUTIONS

1. npr1 mutants are hypersusceptible to TuMV.
1. NIB encoded by TuMV interacts with NPR1.
2. Nib binds the SUMO-interacting motif 3 (SIM3) in the ankyrin repeat domain of NPR1.
3. Nib interferes with NPR1-SUMO3 interaction and suppresses the sumoylation-dependent phosphorylation of NPR1.
4. Binds the SUMO-interacting motif 3 (SIM3) in NPR1 is a property of Nib encoded by several potyviruses.
5. Observations are not limited to Arabidopsis. Soybean-encoded NPR1 is sumoylated by GmSUMO5.

Reply: Thank you very much for the positive decision. We really grateful to the highlights of our findings.

POINTS THAT NEED TO BE CLARIFIED

1. The model in figure 7 needs to be simplified and updated to accurately reflect the data. In its current form, the panel corresponding to potyvirus-infected is contradictory to PR1 expression levels in Fig. 1e and Supplemental Figure 3.

Reply: Thank you very much for this excellent suggestion. We have carefully revised fig. 7 accordingly.

Reviewer #2 (Remarks to the Author):

In this manuscript, the authors attempt to use a series of results to demonstrate that the Turnip mosaic potyvirus (TuMV) uses its RNA-dependent RNA polymerase Nib to interfere with NPR1 sumoylation, resulting in suppression of NPR1 phosphorylation at S11/S15, which in turn weakens the host immune responses. Based on these results, the authors propose that TuMV and other potyviruses deploy NIB to suppress NPR1-mediated resistance by targeting its sumoylation and consider this mechanism “a novel molecular arms race”.

The Nib-SUMO3-NPR1 triangle is certainly interesting. However, the manuscript focused on one side of the coin and might not reveal the true relationship among the three players.

Reply: Thank you very much for the affirmation of our discovery. We performed additional experiments as request to full answer the Nib-SUMO3-NPR1 triangle (see below).

Major concerns

1. Nib is the enzyme the replicates the viral genome. After translation, NIB is transported into the nucleus where it is sumoylated by SUMO3 and subsequently shuttled back to the cytoplasm to replicate the viral genome. Nuclear NPR1 not only competes for SUMO3 but also directly interacts with NIB. It is thus possible that NPR1 also suppresses NIB activity or keeps NIB in the nucleus. This hypothesis is supported by Figures 2d to 2h. Although PR1 was similarly induced in the WT,

NPR1-GFP, sim3-GFP, and *npr1-sim3* plants, NPR1-GFP showed significantly enhanced resistance, whereas *npr1-sim3* appeared to be more susceptible than the wild type (Figures 2g and 2h). If PR1 is used as the marker of immune responses, it can be concluded that it is the NPR1-GFP protein, not immune responses, that suppresses the viral propagation. Is it possible to test if NPR1 protein could suppress NIB activity? The T-DNA insertion line SAIL_708_F09 that accumulates a truncated NPR1 protein containing the ANK repeats (Ding et al., 2020 Front Plant Sci) would be a good material to uncouple NPR1-mediated immune responses and the ANK domain-mediated interaction with NIB. The results of S11/15D in Figure 4 further support the function of NIB-NPR1 interaction in suppressing NIB activity. The authors may want to show TuMV-induced PR1 expression in the lines used in Figure 4. The S11/15D protein has a much stronger interaction with NIB and the S11/15D plants were not infected by the virus. The *sim3/S11/15D* plants are also resistant due to the new interaction created by the S11/15D mutation.

It is possible that the NIB-NPR1 interaction suppresses the activity of both proteins. The authors showed the effect on NPR1; but the effect on NIB activity may play a more important role in the plant-virus interaction.

Reply: Thank you very much for this excellent comment. We obtained the T-DNA insertion line SAIL_708_F09 (*npr1-6*) and directly compared the infectivity of TuMV and the nucleocytoplasmic partitioning of NIB in *npr1-0*, *npr1-1*, and *npr1-6*. Unfortunately, our results did not support the claim that NPR1 directly affect TuMV infection or the nucleocytoplasmic partitioning of NIB; instead, our results clearly support the notion that the NPR1-dependent immune responses affect virus proliferation (Fig. 2; Supplementary Fig. 6-7). Based on these results, we concluded that NPR1 neither suppresses NIB activity nor affects its translocation to the cytoplasm. This new information has been integrated into the revision.

2. Both NIB and NPR1 are conserved components in their own systems. It would be difficult to imagine that the virus and the plant use such important proteins for an arms race. The NIB-NPR1 interaction may be fortuitously formed and may not be a result of evolution under certain selections.

Reply: Thank you very much for this comment. Plant viruses only encode a small number of proteins; as a result, viral proteins are always multifunctional. It is not surprising to find a viral replicase involved in other biological processes, e.g., the replicase of tobacco mosaic virus is a suppressor of RNA silencing (MPMI, 2004 17: 583–592.). Potyviruses adopt a special coding strategy that ten out of eleven proteins are translated from a large polypeptide, which results in most proteins have the same amounts, e.g., the amount of NIB is the same as coat protein (CP). However, one potyviral genome needs more than 1600 copies of CP for encapsulation but was synthesized by one NIB protein. Therefore, it is well accepted that only a small part of NIB in the cytoplasm are directly involved in viral replication, and the rest are “stored” in the nucleus. Therefore, it is not surprising that NIB also has the function of suppressing plant immunity beside the RdRp activity.

On the other hand, NPR1 plays key important role in SA-mediated immunity. Several bacterial and fungi effectors targets NPR1 and suppress its function (discussion part). It is not surprising to finding it was also targeted by viral protein. Indeed, our results clearly NPR1 does not inhibit NIB function but NIB suppress NPR1 function.

3. The results in Figures 2g and 2i are quite different based on the p values, but it is stated that “a similar trend of accumulation of viral RNAs in those transgenic plants”. GFP silencing cannot be excluded.

Reply: Thank you very much for this excellent comment. We performed a deep sequencing analysis of these transgenic seedlings and found that one line of 35S::NPR1-GFP have GFP-derived siRNAs, which explained the observed phenotype. We have included these results and carefully revised the related context accordingly.

Minor comments

1. Page 10 line 237, it may be more likely the SIM2 site is involved in the interaction.
2. Page 12 line 291, the fact that PR1 expression levels are correlated with S11/15D protein levels does not indicate that the phosphorylation is essential.
3. Page 13 line 309, “these results indicate that binding of Nib to SIM3 of NPR1 not only inhibit NPR1-SUMO3 interaction, but also disrupts the phosphorylation of S11/15”. Results in Figures 4a to 4f really indicate that S11/15 phosphorylation enhances the Nib-NPR1 interaction and resistance to the virus. Figure 4g shows reduced phosphorylation.
4. Figure 2b, bright-field images would be helpful. Figure 3a magnification should be increased.
5. Figure 3g, anti-SUMO3 is not in the figure.
6. Page 38, Supplemental fig. 12 legend, d analysis of S11/15D transcript, but in the figure it is PR1.

Reply: Thank you very much for pointing out these mistakes. We have carefully revised accordingly and asked two English natives to correct the manuscript thoroughly.

Reviewer #3 (Remarks to the Author):

In this paper, Liu et al further analysed the role of NPR1 sumoylation on Arabidopsis susceptibility to TuMV. They show that NPR1 is involved in slightly repressing TuMV progression in Arabidopsis but that this repression is slowed down by the action of the TuMV NIB that inhibits NPR1's sumoylation.

It is of interest, but I am not sure I completely grasp how significant this arms race means for the interaction between plant and virus. Besides, I am some technical points that might need clearing. Overall, the paper present an impressive set of experiments but I wonder if the number of repeats or the effectiveness are sufficient to conclude at each step.

Reply: Thank you very much for affirmation of our results.

1- I did not find the starting point very clear (or it is very lucky). (page 5-6 lines 119-122. Was NPR1 the only candidate picked? This result is highlighted by the authors as the most interesting of the study (page 16, line 402), yet from the beginning of the results section it looks like only NPR1 was tested as a potential interactant for NIB ? Is that so ? Wasn't this supported by other data (Y2H screening of a library .. ?)

Reply: Thank you very much for this great comment. We actually cloned all known substrate of SUMO3 and tested their interaction with NIB by BiFC assay. We have included these results and revised the manuscript accordingly.

2- A lot of this paper relies on quite slight differences in virus accumulation. Total accumulation of virus is carried out at 14 dpi, which is quite early. I wonder what happens at longer time after infection, and if the overall increased susceptibility is meaningful over time. Also, I could not find how was tissue sampling carried out for viral and CP quantification (Figure 1 h-i-j). Because at that stage the infection is very unequal in the plant (as seems by GFP imaging in figure 1-f), the sampling can have a very profound effect on the quantification, knowing that the number of samples (6) is quite low. Was the whole plant ground or only selected leaves?

Reply: Thank you very much for these two wonderful comments.

On Arabidopsis, TuMV usually needs 5-6 days to spread from the inoculation leaf to upper non-inoculated leaves and needs about 2-3 weeks to spread to other leaves. At about 4 weeks post inoculation, the plants will senesce and many leaves will die. Therefore, 2-3 weeks post inoculation is the best time for sampling. In our study, the time of sampling (14 or 18 dpi) is the best time to distinguish the susceptibility of the mutants or transgenic plants, which are determined by at least three times of repeats.

Virus can never spread to all cells of a host; therefore, we routinely take the whole aerial part for subsequent analyses to eliminate the influence of sampling on virus accumulation (This information has already been included in some figure legends in the previous edition). We have clearly indicated the detailed sampling methods in the M&M section.

3- Page 9 line 219 and following, figure 2i, a similar trend was observed with 6K2 mCherry: I don't see that on the figure, to me (and statistically), all samples looks like they accumulate the same level of virus.

Reply: Thank you very much for this excellent comment. We performed a deep sequencing analysis of these transgenic seedlings and found that one line of 35S::NPR1-GFP have GFP-derived siRNAs, which explained the observed phenotype. We have included these results and carefully revised the manuscript accordingly.

4- Figure 3-e, is it possible to conclude and make statistics on n=5 nuclei (is the the right n?)? Does it mean only one sample (leaf) was tested per condition?

Reply: Thank you for this comment. As indicated in the method section, all nuclei of a given micrograph are determined by the H2-mRFP signal, then the nuclear YFP signals of each nucleus are calculated, and the average nuclear YFP signal of the micrograph is calculated. The bar chart in Fig. 3e (3a in the revision) data are the mean value of the average nuclear YFP signal of 5 micrographs (more than 200 nuclei). Besides, all experiments in this study including this experiment have been repeated at least three times for accuracy. We have submitted all original micrographs along with our manuscript.

5- Page 11 line 260. « Results show that the sumoylated NPR1 was reduced by about 60% at the presence of Nib. » I am not a specialist but is Fig 3g sufficient to conclude ? How many repeats were done ? I would have the same question for the split-luciferase (Fig3F and 3H)

Reply: Thank you very much for this excellent comment. As indicated in the manuscript, all experiments have been repeated at least three time. We believe that the data in Fig 3 have clearly showed that the sumoylation of NPR1 is influenced by Nib but not the accuracy degree of decrease. We have tone-down the precise reduced level (about 60%) in the revision accordingly.

6- Targeting the SIM3 of NPR1... Which NPR1 is used for interaction studies? Is it the Arabidopsis one. It might be of interest to show how homologs Arabidopsis NPR1 and the one from species that are natural host for the tested viruses (unless they all infect Arabidopsis?) actually, it is in figure 6a ?

Reply: Thank you very much for this interesting comment. The Arabidopsis NPR1 was used in figure 5a (We think that the reviewer meant the 5a but not the 6a). We also tested the interaction between Nib of soybean mosaic virus (SMV) and soybean-encoded NPR1 or its mutant Gmsim3 in Supplementary figure 19. Although not all combinations were tested, our results clearly support the notion that most if not all potyviral Nib able to target host NPR1.

7- Discussion, page 15, lines 366-367. I am not sure whether it is that striking. PR1 induction on TuMV infection is very much reduced in a npr1 KO, but this is accompanied by a very modest

increase in virus load, and quite early on. Therefore, i think it is interesting, but I am not sure it is very significant.

Reply: Thank you very much for this comment. An increase of 2 fold viral genomic RNA results in the almost double leaf areas are infected by TuMV in the same time period (Fig. 2a-b), which may significantly increase the change to be transmitted by aphid. Besides, the transgenic plants expressing S11/15D or sim3|S11/15D showed high levels of resistance to TuMV. These results suggest that NPR1-mediated resistance have an important role in restricting TuMV infection, which can also be manipulated for viral disease management in the future.

Minor points:

Line 187, please rephrase, missing words?

To me, all Y2H experiments should be supported by western blot to ensure that a lack of interaction is not caused by the truncated/mutated proteins being unstable, this is especially true for experiments aiming at analyzing quantitative interactions (Fig 3C, 4D) but I leave it at the editor's discretion

Page 18 line 444, I am not sure what to make of reference 58, where NIB is simply described as an avirulence factor.

Page 18 line 447, word missing (be ?)

Reply: Thank you so much for these comments. We have included all immunoblot results as supplementary figures in the revision and corrected other mistakes accordingly.

Reviewer comments, second round –

REVIEWER COMMENTS

Reviewer #2 (Remarks to the Author):

My major concern was whether the resistance is conferred by NPR1-mediated immune responses or NPR1-mediated suppression of NlB. The authors used the T-DNA insertion line SAIL_708_F09 that accumulates a truncated NPR1 protein to demonstrate that the truncated NPR1 does not provide resistance. Please confirm the truncation is at AA432. If it is at AA432, can NPR1 AA 1-432 interact with NlB? Note that the ANK domain tested in the manuscript is AA231-465.

In the revised manuscript, the authors show that the resistance in 35S::NPR1-GFP-4 is due to silencing of the TuMV-GFP, which is acceptable.

However, the authors did not address my question about immune responses in the lines used in Figure 4. The 35S::S11/15D lines show constitutive PR1 expression and are extremely resistant. PR1 gene induction in these lines may be reduced due to the constitutive expression. The 35S::sim3/S11/15D lines are also resistant. Did they show constitutive PR1 expression? How about PR1 gene induction in these lines in comparison with the wild type and 35S::sim3/S11/15A?

Reviewer #4 (Remarks to the Author):

NPR1 is a star molecule in SA-mediated basal and systemic acquired resistance in plants. In this work, the authors extend their previous studies by demonstrating that TuMV NlB protein directly interacts with the NPR1 and acts as a surrogate to prevent the SUMO3-mediated sumoylation of NPR1, thereby compromising the activation of SA-mediated defense response, they also provide some evidence that NlB affects the phosphorylation of NPR1. In addition, they test the interaction of NlB proteins from diverse potyviruses with the NPR1 and consistently showed NlBs' interference with the NPR1-SUMO3 module. At last, they propose a model regarding the mechanism underlying NlB-SUMO3-NPR1 interactions.

Considering that this work has already been reviewed once at Nature Communications but original reviewer #3. In my opinion, the author has addressed most of the Reviewer #3's concerns. For the tissue sampling at 14 dpi, I agree with the author's response and 14 dpi is a suitable time for harvesting samples, which can distinguish susceptibility or resistance of Arabidopsis to TuMV infection.

Since Reviewer #3 expressed concerns about the issue of sampling timepoint, and the author claimed that "the time of sampling (14 or 18 dpi) is the best time to distinguish the susceptibility of the mutants or transgenic plants". It is recommended to conduct a time-course analysis of TuMV-inoculated wild-type or mutated Arabidopsis plants to better support the feasibility of the author's choice of 14 or 18 dpi as the sampling time.

In addition, I have some other suggestions for the author to consider.

1. Lines 116-122 and Supplementary Fig. 1, Since BiFC shows that NlB interacts with many substrates of SUMO3, how much does the NlB-NPR1 interaction contribute to the immune suppression of NlB? From the infection phenotype presented in Fig. 2a, the loss of NPR1 seems to only moderately affect the infection of TuMV, so it is questionable whether the NlB mainly targets NPR1 to counteract the anti-viral immune responses. At least the author can discuss this.
2. Figure 3c, some important controls are missing, I wonder how about the effect of NlBsim2, NlBk409R and NlBsim2|K409R mutants on the sumoylation of NPR1.
3. I am not convinced by the data shown in the bottom panel of Fig. 4f. Usually, there should be lagging protein bands to indicate the occurrence of phosphorylation when using the Phos-tag

method. However, there are only one horizontal protein band in the bottom panel of Fig. 4f. Please explain why.

4. The conclusion "NIB abolishes the phosphorylation of NPR1 at Ser11/Ser15" is not solid enough. On one hand, NIB only attenuate but not abolish the phosphorylation of NPR1; On the other hand, phosphorylation of NPR1 occurs not only at Ser11/Ser15, but also at Ser55/Ser59 (Saleh et al., 2015, *Cell Host & Microbe* 18, 169–182). The results presented in Fig. 4f lack direct evidence to prove that NIB indeed affects the phosphorylation of NPR1 at Ser11/Ser15 sites, and also, the phosphorylation of Ser55/Ser59 within NPR1 cannot be ruled out based on the Fig. 4f. I suggest modifying these descriptions.

5. Previous studies revealed that phosphorylation of NPR1 at Ser55/Ser59 sites would inhibit its sumoylation. Meanwhile, NPR1 is constantly degraded by the host 26S proteasome (Chen et al., 2017, *Cell Host & Microbe*). The current data cannot rule out the possibility that NIB can also inhibit the NPR1 activity by interfering the phosphorylation of NPR1 at Ser55/Ser59 site or the ubiquitination of NPR1. The author may discuss this in the manuscript.

6. The regulation of NPR1 is tightly controlled by various posttranslational modifications (Saleh et al., 2015, *Cell Host & Microbe*; Chen et al., 2017, *Cell Host & Microbe*). As sumoylation facilitates proteasome-mediated degradation of NPR1, if NIB indeed inhibits sumoylation of NPR1, why the accumulation level of NPR1 protein does not change significantly in Fig. 3c and Fig. 4f (Fig. 4f, panel anti-NPR1, lane 3 Vs lane 5)? I suggest the authors testing the stability of NPR1 in the presence of NIB or its derivatives (eg. NIBsim2, NIBK409R and NIBsim2|K409R) by using in vitro degradation experiments. I think this would strengthen the conclusion of this study.

7. In most cases, the authors examine the effect of NIB on the sumoylation and phosphorylation by overexpression of NIB protein. Changes in the sumoylation and phosphorylation levels of NPR1 during TuMV infection could be analyzed to better reflect the NPR1 functions in the context of virus infection.

Point-to-point response to reviewer comments

Reviewer #2 (Remarks to the Author):

My major concern was whether the resistance is conferred by NPR1-mediated immune responses or NPR1-mediated suppression of Nlb. The authors used the T-DNA insertion line SAIL_708_F09 that accumulates a truncated NPR1 protein to demonstrate that the truncated NPR1 does not provide resistance. Please confirm the truncation is at AA432. If it is at AA432, can NPR1 AA 1-432 interact with Nlb? Note that the ANK domain tested in the manuscript is AA231-465.

Response: This is a very thoughtful and interesting comment. We confirmed that the T-DNA was inserted at 466 aa (1397 nt) in SAIL_708_F09 by Sanger sequencing (ABI peak file has been uploaded) and then analyzed the subcellular localization of the truncated NPR1 (NPR1 Δ C) and its interaction with Nlb by BiFC, Y2H, and Co-IP. Our results have shown that NPR1 Δ is located mostly in the cytoplasm and interacted with Nlb with similar intensity as wild-type NPR1. These new results further reinforced the notion that the resistance is conferred by NPR1-mediated immune responses but not by NPR1-mediated suppression of Nlb.

In the revised manuscript, the authors show that the resistance in 35S::NPR1-GFP-4 is due to silencing of the TuMV-GFP, which is acceptable.

However, the authors did not address my question about immune responses in the lines used in Figure 4. The 35S::S11/15D lines show constitutive PR1 expression and are extremely resistant. PR1 gene induction in these lines may be reduced due to the constitutive expression. The 35S::sim3/S11/15D lines are also resistant. Did they show constitutive PR1 expression? How about PR1 gene induction in these lines in comparison with the wild type and 35S::sim3/S11/15A?

Response: Thank you very much for this excellent and thought-provoking comment. We carefully determined the expression levels of *PR1* in these transgenic plants before and after TuMV infection to get a detailed picture of the resistance and immune-inducing ability of these transgenic plants. As expected by Reviewer #2, our results showed that TuMV infection did not further induce the expression of *PR1* in the two 35S::S11/15D lines, which supports the notion that the extreme resistance of 35S::S11/15D was caused by the constitutive expression of *PR* genes; Both lines of 35S::sim3/S11/15A had a similar expression level of *PR1* as WT plants under steady-state conditions, but TuMV infection could only slightly induce the expression of *PR1* in

35S::sim3/S11/15A, which explained the hyper-susceptible phenotype of 35S::sim3/S11/15A. Surprisingly, the expression levels of *PR1* in both lines of 35S::sim3/S11/15D were similar as WT plants under steady-state conditions, but were elevated to a comparable level as virus-infected wild-type plants. Given the resistant phenotype of both lines of 35S::sim3/S11/15D (Fig. 4d-f) and the fact that sim3/S11/15D is located mostly in the cytoplasm (Fig. 4F in Saleh et al. Cell Host & Microbe 18, 169–182) and, we suspected that sim3/S11/15D is functional but is still located in the cytoplasm as multimers and the infection of TuMV triggers its monomerization and NPR1 monomers enter the nucleus to induce the expression of *PR* genes, e.g., *PR1*. To confirm this hypothesis, we analyzed the amount of sim3/S11/15D-GFP in the supernatant (cytoplasmic fraction) and pellets (nuclear fraction) of cell lysates from mock or SA treated seedlings of 35S::sim3/S11/15D-GFP. We found that SA treatment caused the increment of sim3/S11/15D in the nucleus (Supplementary Fig. 15d). These results further strengthen the notion that phosphorylation at S11/15D is required for the function of NPR1 and is dependent on sumoylation. We have included these results and carefully revised the related context accordingly.

Reviewer #4 (Remarks to the Author):

NPR1 is a star molecule in SA-mediated basal and systemic acquired resistance in plants. In this work, the authors extend their previous studies by demonstrating that TuMV NIB protein directly interacts with the NPR1 and acts as a surrogate to prevent the SUMO3-mediated sumoylation of NPR1, thereby compromising the activation of SA-mediated defense response, they also provide some evidence that NIB affects the phosphorylation of NPR1. In addition, they test the interaction of NIB proteins from diverse potyviruses with the NPR1 and consistently showed NIBs' interference with the NPR1-SUMO3 module. At last, they propose a model regarding the mechanism underlying NIB-SUMO3-NPR1 interactions. Considering that this work has already been reviewed once at Nature Communications but original reviewer #3. In my opinion, the author has addressed most of the Reviewer #3's concerns.

Response: Thanks to Reviewer 4 for a very positive review of our manuscript and the affirmation of our modifications.

For the tissue sampling at 14 dpi, I agree with the author's response and 14 dpi is a suitable time for harvesting samples, which can distinguish susceptibility or resistance of Arabidopsis to TuMV infection. Since Reviewer #3 expressed concerns about the issue of sampling timepoint, and the author claimed that "the time of sampling (14 or 18 dpi) is the best time to distinguish the susceptibility of the mutants or transgenic plants". It is recommended to conduct a time-course analysis of TuMV-inoculated wild-type or mutated Arabidopsis plants to better support the feasibility of the author's choice of 14 or 18 dpi as the sampling time.

Response: Thanks for pointing this out. Since sampling at 14-18 dpi is widely accepted by most researchers and it is not the major subject of this study, we do not think it is necessary to redo a new time-course analysis. Nevertheless, we uploaded the phenotype of seedlings of WT, *npr1-1*, and

npr1-0 infected by TuMV at 14 dpi and 30 dpi as materials for review purpose.

In addition, I have some other suggestions for the author to consider.

1. Lines 116-122 and Supplementary Fig. 1, Since BiFC shows that Nib interacts with many substrates of SUMO3, how much does the Nib-NPR1 interaction contribute to the immune suppression of Nib? From the infection phenotype presented in Fig. 2a, the loss of NPR1 seems to only moderately affect the infection of TuMV, so it is questionable whether the Nib mainly targets NPR1 to counteract the anti-viral immune responses. At least the author can discuss this.

Response: Thank you very much for this interesting comment. We totally agree with this notion that other SUMO3 substrates including those we mentioned in this study may also contribute to antiviral immunity. From the expression of *PR1* in *npr1* mutants after TuMV infection, it is clear that the induction of *PR1* expression is largely dependent on NPR1 and obviously there are other factors that also contribute to the expression of *PR1* (Fig. 1f and Supplementary Fig. 7). We have discussed these possibilities in the revised manuscript.

2. Figure 3c, some important controls are missing, I wonder how about the effect of Nibsim2, NibK409R and Nibsim2|K409R mutants on the sumoylation of NPR1.

Response: Thank you for this comment. Compared to Nib, its mutants have a reduced ability to interact with NPR1 (Fig. 3a-b; Supplementary Fig.8 and 9); Therefore, it can be predicted that their ability to inhibit sumoylation of NPR1 is also weaker than that of wild-type Nib, or even not at all. The experiment in Fig. 3C is performed with homologous *XVE::Nib 35S::NPR1-GFP* seedlings. Given the fact that we unable to obtain homologous transgenic plants of *XVE::Nibsim2 35S::NPR1-GFP*, *XVE::Nib409R 35S::NPR1-GFP*, and *XVE::Nibsim2/409R 35S::NPR1-GFP* in such a short time, we decided only to discuss this possibility in the revision.

3. I am not convinced by the data shown in the bottom panel of Fig. 4f. Usually, there should be lagging protein bands to indicate the occurrence of phosphorylation when using the Phos-tag method. However, there are only one horizontal protein band in the bottom panel of Fig. 4f. Please explain why.

Response: Thank you for pointing this out. As we mentioned in the Material &Method section, the product we used is called Phosbind Biotin BTL-105/Phos binding reagent (Phosbind) Biotin LC (cat no. F4004), which is not the one reviewer #4 mentioned: Phosbind Acrylamide (cat no. F4002). Phosbind Biotin BTL-105 needs streptavidin-conjugated horseradish (HRP) and chemiluminescent detection reagent for detection on a PVDF membrane, while Phosbind Acrylamide is a specific reagent for separation of phosphorylated proteins using SDS-PAGE, which usually result in a lagging band. The differences and detailed instructions can be obtained from the official website of APEX BIO at <https://www.apexbt.com/phos-binding-reagent-biotin-lc.html> and <https://www.apexbt.com/phos-binding-reagent-acrylamide.html>, respectively. We have included this formation in the revision.

4. The conclusion “Nib abolishes the phosphorylation of NPR1 at Ser11/Ser15” is not solid enough. On one hand, Nib only attenuate but not abolish the phosphorylation of NPR1; On the other hand, phosphorylation of NPR1 occurs not only at Ser11/Ser15, but also at Ser55/Ser59 (Saleh et al., 2015, Cell Host & Microbe 18, 169–182). The results presented in Fig. 4f lack direct evidence to prove that Nib indeed affects the phosphorylation of NPR1 at Ser11/Ser15 sites, and also, the phosphorylation of Ser55/Ser59 within NPR1 cannot be ruled out based on the Fig. 4f. I suggest modifying these descriptions.

Response: Thank you very much for this comment. Comparing the phosphorylation levels between

NPR1 and S11/15A (lane 2 and 1 of Fig. 4f), it is clear that phosphorylation mostly taken place at Ser11/Ser15 under our experimental conditions. We believe that the data in Fig 4f have clearly shown that the phosphorylation of NPR1 at Ser11/Ser15 is attenuated by Nib. We have toned down our claim and revised the descriptions accordingly.

5. Previous studies revealed that phosphorylation of NPR1 at Ser55/Ser59 sites would inhibit its sumoylation. Meanwhile, NPR1 is constantly degraded by the host 26S proteasome (Chen et al., 2017, Cell Host & Microbe). The current data cannot rule out the possibility that Nib can also inhibit the NPR1 activity by interfering the phosphorylation of NPR1 at Ser55/Ser59 site or the ubiquitination of NPR1. The author may discuss this in the manuscript.

Response: Thank you for this interesting comment. Based on the latest study (Zavaliev et al. 2020, Cell, 182, 1093-1108), it is believed that phosphorylation of Ser55/Ser59 prohibits the entry of NPR1 into the nucleus and the entry of NPR1 into the nucleus requires the dephosphorylation of Ser55/Ser59. It is believed that sumoylation takes place in the nucleus; thus, the phenomenon that phosphorylation at Ser55/Ser59 inhibits sumoylation is because NPR1 does not enter the nucleus. Based on the phosphorylation levels between NPR1 and S11/15A (lane 2 and 1 of Fig. 4f), it is clear that phosphorylation mostly taken place at Ser11/Ser15 under our experimental conditions. Further considering the factor that Nib does not inhibits the entry of NPR1 into nucleus (Supplementary figure 4g), we concluded that Nib attenuates the phosphorylation of NPR1 at Ser11/Ser15. An in vitro degradation experiment was performed to analyze the influence of Nib on the stability of NPR1 (see below).

6. The regulation of NPR1 is tightly controlled by various posttranslational modifications (Saleh et al., 2015, Cell Host & Microbe; Chen et al., 2017, Cell Host & Microbe). As sumoylation facilitates proteasome-mediated degradation of NPR1, if Nib indeed inhibits sumoylation of NPR1, why the accumulation level of NPR1 protein does not change significantly in Fig. 3c and Fig. 4f (Fig. 4f, panel anti-NPR1, lane 3 vs lane 5)? I suggest the authors testing the stability of NPR1 in the presence of Nib or its derivatives (eg. Nibsim2, NibK409R and Nibsim2|K409R) by using in vitro degradation experiments. I think this would strengthen the conclusion of this study.

Response: Thank you very much for this comment. The experiments in Fig. 3c and Fig. 4f were performed with affinity-purified NPR1. We deliberately loaded the same amount of NPR1 or its mutants to analyze the effects of Nib or its mutants on the sumoylation or phosphorylation of NPR1. As requested, we performed an in vitro degradation experiment to test the influence of Nib on the stability of NPR1 according to Spoel et al. (Cell, 2009, 137, 860-872). Our results have shown that the degradation ratio of NPR1 is not significantly changed in the presence of Nib, Nibsim2, NibK409R, or Nibsim2|K409R. Thus, we conclude that Nib has no obvious influence on the stability of NPR1. These data have been included as materials for review purpose but not for publication.

7. In most cases, the authors examine the effect of Nib on the sumoylation and phosphorylation by overexpression of Nib protein. Changes in the sumoylation and phosphorylation levels of NPR1 during TuMV infection could be analyzed to better reflect the NPR1 functions in the context of virus infection.

Response: Thank you very much for this thoughtful comment. We must keep in mind that Nib is the RdRp of potyviruses; as a result, any change may have a serious impact on its RdRp activity, or even cause a complete loss of its RdRp activity. As indicated in our previous study (Plant cell, 2017, 29, 508-525), TuMV infectious clone which harbors the Nibsim2 mutant is lethal. Therefore, it is

impossible to analysis the sumoylation or phosphorylation of NPR1 under the context of virus infection with suitable controls.

Reviewer comments, third round –

REVIEWER COMMENTS

Reviewer #2 (Remarks to the Author):

The authors successfully addressed my concerns and I don't have further questions and comments.

Reviewer #4 (Remarks to the Author):

During the long-term arms race between pathogens and plants, plants have evolved a series of disease resistance genes, such as NPR1 studied in this work, to defend against pathogen infection. However, pathogens have also evolved corresponding counter-defensive strategies to suppress plant immunity, such as the potyviruses NIB protein targets NPR1 to achieve persistent infection as described in this work. In the revised manuscript, Liu et al. have addressed most of my concerns raised in my previous review, although I am not entirely satisfied with their response to some points. For example, the below points need to be clarified before it can be acceptable for publication

1. The genetic evidence in this study is not strong enough. In response to Point #2, the authors did not obtain transgenic Arabidopsis plants that co-express different proteins, which somewhat weakens the genetic basis of this study. As a plant research model, genetic manipulation for Arabidopsis is simple and various genetic materials are easily obtained in comparison to other plants, which are difficult to transform and have longer growth period. The lack of some transgenic materials as the control in this work weakens the conclusions to some extent.
2. In the author's response to point 6, if the authors had the genetic materials as mentioned above, it would be more convenient and reliable to demonstrate whether NIB affects the stability of NPR1. The authors preliminarily tested the effects of various NIB mutants on NPR1 stability, and surprisingly, NIB and its mutants had little effect on NPR1 stability (In fact, the author's quantification of protein bands is not accurate). This seems to contradict the logic of the previously reports showing that sumoylation facilitates proteasome-mediated degradation of NPR1. The author should explain this.
3. In the author's response to point 7, I understand that as an RdRp of potyviruses, mutations in NIB are often lethal for the virus. However, I wonder whether TuMV infection indeed affects the sumoylation and phosphorylation levels of NPR1 in comparison to mock-inoculated plants. Has the author performed any experiments on this?

Point-to-point response to reviewer comments

Reviewer #2 (Remarks to the Author):

The authors successfully addressed my concerns and I don't have further questions and comments.

Response: We really appreciate your affirmation on our results and efforts.

Reviewer #4 (Remarks to the Author):

During the long-term arms race between pathogens and plants, plants have evolved a series of disease resistance genes, such as NPR1 studied in this work, to defend against pathogen infection. However, pathogens have also evolved corresponding counter-defensive strategies to suppress plant immunity, such as the potyviruses NlB protein targets NPR1 to achieve persistent infection as described in this work. In the revised manuscript, Liu et al. have addressed most of my concerns raised in my previous review, although I am not entirely satisfied with their response to some points. For example, the below points need to be clarified before it can be acceptable for publication.

1. The genetic evidence in this study is not strong enough. In response to Point #2, the authors did not obtain transgenic Arabidopsis plants that co-express different proteins, which somewhat weakens the genetic basis of this study. As a plant research model, genetic manipulation for Arabidopsis is simple and various genetic materials are easily obtained in comparison to other plants, which are difficult to transform and have longer growth period. The lack of some transgenic materials as the control in this work weakens the conclusions to some extent.

Response: Thank you again for this comment. As we addressed in the previous communication, NlB mutants have a reduced ability to interact with NPR1 (Fig. 3a-b; Supplementary Fig.8 and 9); thus, it is apparent that NlB mutants should have weaker, or even no, ability to inhibit the sumoylation of NPR1. To avoid the influence of mutation on NlB activity, we have produced adequately transgenic plants of NPR1 and its mutants and detailed analyzed their antiviral capacity. Our results showed that the transgenic seedlings of sim3|S11/15D but not wild-type NPR1 have a high level of resistance to TuMV, which supports the notion that NlB influences the sumoylation and subsequent phosphorylation of NPR1. Therefore, we believe that our conclusions are solid supported at both biochemical and genetic levels.

2. In the author's response to point 6, if the authors had the genetic materials as mentioned above, it would be more convenient and reliable to demonstrate whether NlB affects the stability of NPR1. The authors preliminarily tested the effects of various NlB mutants on NPR1 stability, and surprisingly, NlB and its mutants had little effect on NPR1 stability (In fact, the author's quantification of protein bands is not accurate). This seems to contradict the logic of the previously reports showing that sumoylation facilitates proteasome-mediated degradation of NPR1. The author should explain this.

Response: Thank you for this thoughtful comment. We were also surprised by the results of the in vitro degradation experiment. After carefully consideration, we believe that NlB maybe also facilitate degradation of NPR1 after binding to its SIM3 motif since plenty of viral proteins, e.g., P0 of poleroviruses, betaC1 of geminiviral betasatellites, ANK proteins of poxviruses, and E6/E7 or HPV, can function as ubiquitin-protein E3 ligase or utilizes host ubiquitin-26S proteasome to target host factor for degradation. Thus, nuclear NlB-YFP was purified and analyzed by mass spectrometry.

Interestingly, one ubiquitin-conjugating E2 enzyme (UEV1A) and two ubiquitin-protein E3 ligases (PRT6 and DURF2) were identified as NIB-interacting candidates (the MS data has been uploaded). We further confirmed the interaction between NIB and UEV1A by BiFC and Y2H assays. Thus, it is very likely that NIB also utilizes host ubiquitin-26S proteasome to degrade NPR1. We are working hard to explore this interaction at present.

3. In the author's response to point 7, I understand that as an RdRp of potyviruses, mutations in NIB are often lethal for the virus. However, I wonder whether TuMV infection indeed affects the sumoylation and phosphorylation levels of NPR1 in comparison to mock-inoculated plants. Has the author performed any experiments on this?

Response: Thank you very much for this interesting comment. Actually, we have already compared the level of sumoylation and phosphorylation of NPR1 before and after TuMV infection as preliminary experiments during the research. We have included these data in the revision.